# On Robustness of Vision-Language-Action Model against Multi-Modal Perturbations

**Jianing Guo[1, 4], Zhenhong Wu[1], Chang Tu[3], Yiyao Ma[3], Xiangqi Kong[1], Zhiqian Liu[1],**

**Jiaming Ji[2], Shuning Zhang[5], Yuanpei Chen[2, 4], Kai Chen[3], Qi Dou[3], Yaodong Yang[2, 4],**

**Xianglong Liu[1, 6, 7], Huijie Zhao[1], Weifeng Lv[1], Simin Li[1, 3] [*]**

[1]Beihang University, [2]Peking University, [3]The Chinese University of Hong Kong, [4]PKU-Psibot Lab, [5]Tsinghua University, [6]Zhongguancun Laboratory, [7]Hefei Comprehensive National Science Center

## Abstract

In Vision–Language–Action (VLA) models, robustness to real-world perturbations is critical for deployment. Existing methods target simple visual disturbances, overlooking the broader multi-modal perturbations that arise in actions, instructions, environments, and observations. Here, we first evaluate the robustness of mainstream VLAs under 17 perturbations across four modalities. We find (1) actions as the most fragile modality, (2) Existing visual-robust VLA do not gain robustness in other modality, and (3) $\pi_0$ demonstrates superior robustness. To build multi-modal robust VLAs, we propose RobustVLA against perturbations in VLA inputs and outputs. For output robustness, we perform offline robust optimization against worst-case action noise that maximizes mismatch in flow matching objective. This can be seen as adversarial training, label smoothing, and outlier penalization. For input robustness, we enforce consistent actions across input variations that preserve task semantics. To account for multiple perturbations, we formulate robustness as a multi-armed bandit problem and apply an upper confidence bound algorithm to automatically identify the most harmful noise. Experiments on LIBERO demonstrate our RobustVLA delivers absolute gains over baselines of 12.6% on the $\pi_0$ backbone and 10.4% on the OpenVLA backbone across all 17 perturbations, achieving 50.6x faster inference than existing visual-robust BYOVLA that requires external LLMs, and a 10.4% gain under mixed perturbations. On the real-world FR5 robot, under four types of multimodal perturbations, RobustVLA shows strong low-data performance, outperforming $\pi_0$ by 65.6% success rate with 25 demonstrations. Even with abundant demos, our method still outperform $\pi_0$ by 30% success rate. Code and demo videos available at `https://github.com/gakakulicc/RobustVLA`.

## 1 Introduction

Vision–Language–Action (VLA) models are a class of robotic foundation models that enable flexible, general, and dexterous manipulation through vision–language inputs (Zhong et al., 2025; Sapkota et al., 2025). Trained on diverse, internet-scale robot data, VLAs can perform cross-embodied, general-purpose control in real-world settings (Kim et al., 2025; Black et al., 2024; Bjorck et al., 2025). Despite these advances, VLAs remain vulnerable to a wide range of multi-modal uncertainties encountered in practice, including those in observation (*e.g.*, sensory noise, camera errors), action (*e.g.*, sensorimotor noise, unexpected disturbances), environment (*e.g.*, external forces, distracting objects), and language (*e.g.*, synonymous or ambiguous instructions).

Recent work has begun to explore the robustness of VLAs, but efforts remain limited in scope. VLATest (Wang et al., 2025) primarily evaluates VLA robustness against visual perturbations, focusing on uncertainties in environment transitions and observations. For enhancing robustness,

---
[*]Corresponding Author. E-mails: lisiminsimon@buaa.edu.cn.

Figure 1: Framework of our paper. We evaluate VLA robustness under 17 uncertainties across 4 modalities. Based on the findings, we enhance robustness against both VLA inputs and outputs.

BYOVLA (Hancock et al., 2025) mitigates irrelevant visual details by identifying, segmenting, and inpainting them using large vision–language models, while GEVRM (Zhang et al., 2025) improves robustness to common visual corruptions such as color jitter through model-based planning. However, these methods are restricted to visual robustness, leaving their effectiveness against multi-modal uncertainties untested. Moreover, both approaches rely heavily on external large models, leading to substantial computational overhead.

To better understand the robustness of VLAs beyond visual uncertainties, as shown in Fig. 1, we evaluate and enhance the robustness of VLAs to multi-modal perturbations. We begin by evaluating the robustness of mainstream VLAs against 17 uncertainties across four modalities. Our findings are threefold: (1) action is the most fragile modality, (2) existing visual-robust VLAs do not show improvements in other modalities, and (3) $\pi_0$ (Black et al., 2024) demonstrates superior robustness, outperforming OpenVLA (Kim et al., 2025) and $\pi_0$-FAST (Pertsch et al., 2025) by large margins. Based on these results, we recommend that robust VLAs focus on robustness in all modalities and build on $\pi_0$ backbone as a starting point.

Building on evaluation results, we propose RobustVLA, which handles multi-modal uncertainties in both VLA inputs and outputs. RobustVLA is based on the $\pi_0$ backbone and generalizes naturally to other VLAs. For robustness against VLA output, we derive the worst-case action deviation from the flow-matching objective, then match the action head with both the original and worst-case action distributions. This process can be seen as a combination of flow matching with noisy action distributions, label smoothing, and outlier penalization. For robustness against VLA inputs, we ensure that the noise does not alter the semantics of the current state, so the optimal action remains invariant. We thus regularize the objective to maintain consistent output actions across diverse input perturbations. To balance various types of perturbations, we frame the problem as a multi-armed bandit and employ the upper confidence bound (UCB) algorithm (Auer, 2002) to select the most harmful perturbation for training. On LIBERO benchmark, RobustVLA achieves absolute gains of 12.6% on the $\pi_0$ backbone and 10.4% on the OpenVLA backbone across 17 perturbations, being 50.6x faster than visual-robust BYOVLA that requires external LLMs, and achieving a 10.4% gain under mixed perturbations. In real-world deployment with four multimodal perturbations, RobustVLA performs strongly in the low-data regime, outperforming $\pi_0$ by $65.6\%$ with 25 demos, and still achieves 30% higher success rate than $\pi_0$ with 100 demos, where the performance of $\pi_0$ saturates.

**Contributions.** Our contributions are twofold. First, we evaluate the robustness of VLAs under various multi-modal noise and offer suggestions for improving robustness of VLAs. Second, we propose RobustVLA against input and output noise perturbations, which delivers robust gains across 4 modalities and 2 backbones in both simulation and real-world settings.

## 2 RELATED WORK

**Vision-Language-Action (VLA) Foundation Models**. Vision-Language-Action (VLA) models serve as foundational systems for robotics, integrating vision, language, and control. Recent approaches can be categorized into two primary types. Autoregressive VLAs leverage large pretrained VLMs and generate discrete action tokens in an autoregressive manner (Brohan et al., 2022; Zitkovich et al., 2023; O'Neill et al., 2024; Kim et al., 2025; Pertsch et al., 2025; Qu et al., 2025). These action tokens are then decoded into low-level, executable actions. In contrast, diffusion-based VLAs

generate continuous, high-frequency, multi-modal action distributions by outputting action sequences through a diffusion-based action head (Team et al., 2024; Li et al., 2024; Black et al., 2024; Bjorck et al., 2025). While these models excel in general-purpose embodied decision-making tasks, their robustness remains a significant concern. Existing research on robust VLAs primarily focus on visual input. For instance, VLATest (Wang et al., 2025) demonstrates that current VLAs are vulnerable to various visual corruptions. To mitigate visual perturbations, BYOVLA (Hancock et al., 2025) removes model-sensitive features in visual inputs using VLM-based segmentation and inpainting, while GEVRM (Zhang et al., 2025) addresses common visual corruptions, such as color jitter, through model-based planning. However, these robust VLA methods focus solely on visual inputs and fail to account for multi-modal uncertainties in real-world. Furthermore, all of these approaches rely extensive assess to external large models, leading to substantial computational overhead.

**Robust Decision Making**. Before the advent of VLAs, robust decision-making was primarily explored within the framework of RL, specifically through robust MDPs (Nilim & El Ghaoui, 2005; Iyengar, 2005). Uncertainties in these settings can arise from various components of MDPs, including environment transitions (Pinto et al., 2017; Mankowitz et al., 2019), actions (Tessler et al., 2019), states (Zhang et al., 2020; 2021), and rewards (Wang et al., 2020). In environments with simulators available, RL agents can learn robust policies through minimax optimization against worst-case adversaries. However, in the case of VLAs, only offline datasets are available, akin to the settings of behavior cloning (Schaal, 1996) and offline RL (Levine et al., 2020). Achieving robust decision-making in the absence of an interactive environment is more challenging, as policies under uncertainty may lead to actions outside the distribution of the original dataset, causing OOD transitions. Consequently, robustness is typically achieved in such settings for states (Shen et al., 2020; Yang et al., 2022; Rigter et al., 2022) and environment transitions (Panaganti et al., 2022; 2023; Seo et al., 2024), with the goal of retaining the original policy despite deviations. However, two major challenges remain when applying these techniques to VLAs. First, it remains unclear how to robustly handle a diverse range of perturbations, with existing methods achieving robustness only against a limited set of environmental uncertainties (Agrawal et al., 2023). Second, it is yet unknown how to achieve action-robust offline RL, as OOD transitions are inevitable in real-world scenarios.

## 3 EVALUATING THE ROBUSTNESS OF VLAS

In this section, we evaluate the robustness of mainstream VLAs by first presenting the problem formulation, then detailing the experimental setup, and finally summarizing the main findings.

**Problem Formulation.** We model the decision process of VLAs as a Partially Observable Markov Decision Process (POMDP) (Kaelbling et al., 1998), defined as a tuple $G = \langle \Psi, \mathcal{S}, \mathcal{O}, O, \mathcal{A}, \mathcal{P}, R, \gamma \rangle$. Here, $\Psi$ is the space of language instructions, $\mathcal{S}$ is the state space, $\mathcal{O}$ is the observation space, $O$ is the observation emission function, $\mathcal{A}$ is the action space, $\mathcal{P} : \mathcal{S} \times \mathcal{A} \to \Delta(\mathcal{S})$ is the transition function, $R : \mathcal{S} \times \mathcal{A} \times \Psi \to \mathbb{R}$ is the reward function.

We follow a practical formulation similar to $\pi_0$ (Black et al., 2024). At $t = 0$, a language instruction $\psi \in \Psi$ was given. At time $t$, the robot operates in state $s_t \in \mathcal{S}$, observing 2-3 RGB images and the language instruction $\mathbf{o}_t = \{o_t^1, ...o_t^n, \psi\} = O(s_t)$. The robot takes an action chunk $A_t = [a_t, a_{t+1}, ...a_{t+H}]$ according to its policy $\pi(A_t|\mathbf{o}_t)$. The policy takes observation as input and partial observability is implicitly encoded via historical input in VLMs. The environment proceeds to $s_{t+1} \sim \mathcal{P}(\cdot|s_t, A_t)$ and receive reward $r_t = R(s_t, A_t, \psi)$. $\gamma \in [0, 1)$ is the discount factor.

**Robustness under uncertainties.** We define test-time uncertainties for VLAs as $\omega \in \Omega$, with $\Omega = \{\Omega_\psi \subseteq \Psi, \Omega_o \subseteq \mathcal{O}, \Omega_a \subseteq \mathcal{A}, \Omega_p \subseteq \mathcal{P}\}$ the uncertainty set that perturbs instructions, observations, actions and environments, respectively. Given a task $\psi$, the robustness of VLAs is defined as:

$$J^{\text{robust}}(\pi, \psi) = \mathbb{E}_{\omega \sim \Omega} \Big[ \mathbb{E}_{s_0 \sim \rho_0} \mathbb{E}_{\pi, \omega} \Big[ \sum_{t=0}^{\infty} \gamma^t r_t \,|\, s_0, \psi \Big] \Big]. \tag{1}$$

### 3.1 EXPERIMENT SETTINGS

**Evaluated Algorithms.** We consider OpenVLA (Kim et al., 2025), $\pi_0$-FAST (Pertsch et al., 2025) and $\pi_0$ (Black et al., 2024) as representative of autoregressive and diffusion-based VLAs, using their publicly released checkpoint. For robust VLA methods, we consider BYOVLA (Hancock et al., 2025)

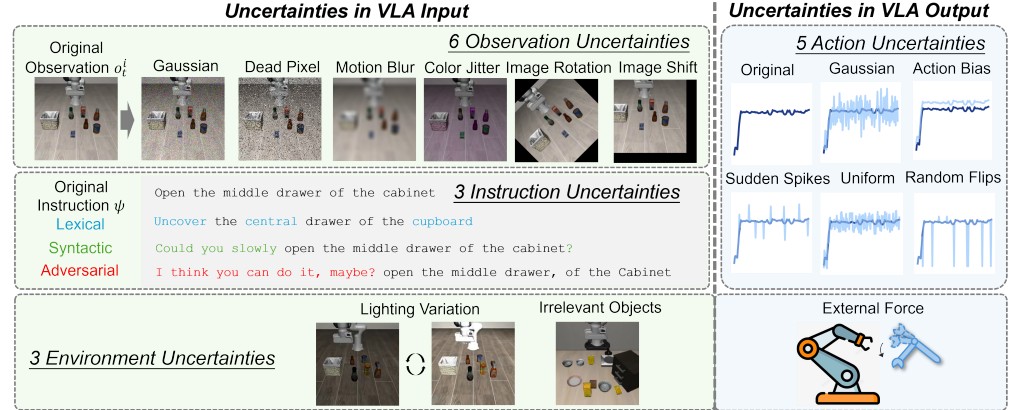

Figure 2: Overview of 17 uncertainty types spanning observation, environment, instruction, and action modalities, used in our evaluation of VLA robustness.

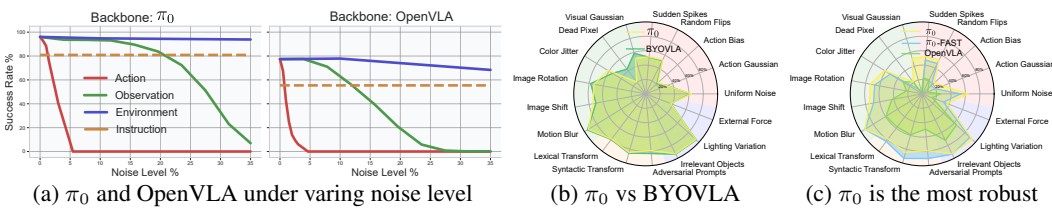

(a) $\pi_0$ and OpenVLA under varing noise level      (b) $\pi_0$ vs BYOVLA      (c) $\pi_0$ is the most robust

Figure 3: Selected results of robustness evaluation. Numerical results available in Section. 5.

on $\pi_0$ backbone , we use the official implementation and apply its robustness enhancement procedure only to the visual modality. Due to the lack of publicly available code and insufficient implementation details, we were unable to reproduce GEVRM (Zhang et al., 2025) for fair comparison.

**Evaluated Uncertainties.** We consider 17 perturbations in action, observation, environment and instructions, see Fig. 2 for visualization. **Action uncertainties** capture sensorimotor noise, actuator wear, and unexpected disturbances in VLA output, modeled by 5 types: uniform noise, Gaussian noise, action bias, random flips and sudden spikes. **Observation uncertainties** arise from sensory noise and camera error that affect VLA input, with 6 variants: Gaussian noise, dead pixel, motion blur, color jitter, image rotation and image shift. **Environment uncertainties** contains 3 types of external influences, with external force in VLA output, irrelevant objects and lighting variations in VLA input. **Instruction uncertainties** represent 3 linguistic variability in VLA input, including lexical transformations at the word level, syntactic transformations at the sentence level, and adversarial prompts with ambiguity and distraction. See details in Appendix. A.

**Evaluation Process.** For robustness evaluation, we apply each uncertainties to our evaluated algorithms. We use LIBERO benchmark suite (Liu et al., 2023) for evaluation, following their official settings. Due to space limit, we report crucial experiment results that support our findings in this section and leave numerical results to Section. 5.

## 3.2 EXPERIMENT RESULTS

We present our key findings below with corresponding experiment results.

**Action is the most fragile modality.** We first evaluate the robustness of all modalities with varying noise level. We unify noise level for all modalities as the percentage of the maximum allowable value in each modality. As shown in Fig. 3a, as the magnitude of noise increases, the robustness of action drops drastically. For example, the success rate of $\pi_0$ are reduced to 52.4% under noise 0.05 (2.5%), and fails completely at 0.1 (5%). This is in stark contrast compared with action-robust RL, where noise are often set to 0.1 or higher (Tessler et al., 2019). An explanation from offline RL theory (Levine et al., 2020) is that VLA policies trained on fixed datasets are especially vulnerable to action errors: a single mistake can drive the rollout off-distribution, causing errors to accumulate quadratically with the horizon, whereas in online RL the accumulation is only linear since interaction allows partial correction. In contrast, we find VLAs more robust against uncertainties in VLA input, while non-robust against observations only at high uncertainties.

**Existing visual-robust VLAs do not show improvements in other modalities.** Next, we investigate the robustness of BYOVLA, an off-the-shelf visual-robust VLA that identify and replace the sensitive regions in visual input. As shown in Fig. 3b, BYOVLA improves performance under Gaussian noise by 7.3% and dead pixels by 22.3%, yet the average gain across all visual corruptions is only 4.0%. Moreover, we observe no measurable improvement ($+0.0\%$) in non-visual modalities. These results indicate that visual robustness alone does not translate into broader robustness across other modalities, motivating our study of comprehensive multi-modal robustness in VLA models.

$\pi_0$ **demonstrates higher robustness than OpenVLA and $\pi_0$-FAST.** As shown in Fig. 3c, $\pi_0$ outperforms OpenVLA and $\pi_0$-FAST by 27.9% and 5.1%, respectively. Since $\pi_0$ and $\pi_0$-FAST share the same VLM backbone, this difference suggests an advantage of the diffusion-based action head. We evaluate both models under varying noise and observe that the robustness of $\pi_0$-FAST degrades faster than $\pi_0$ (Appendix C.2), which motivates us to develop RobustVLA on the $\pi_0$ backbone.

## 4 ROBUSTVLA

To counter multi-modal uncertainties, we propose RobustVLA, a fine-tuning framework to enhance robustness against uncertainties in VLA inputs and outputs. We take $\pi_0$ with rectified flow matching as an example due to its popularity, while our method naturally extends to VLAs with general diffusion-based action head and autoregressive VLAs like OpenVLA.

### 4.1 ROBUSTNESS AGAINST VLA OUTPUTS

Ensuring robustness against action output can be hard, since we are equipped with an offline dataset, and any action noises inevitably make subsequent transitions OOD, which amplifies subsequent errors (Levine et al., 2020). In our paper, taking $\pi_0$ with a rectified flow matching action head, we first derive $\ell_p$ bounded worst-case action noise by maximizing the flow matching loss, then performing robust optimization against such noise.

**Preliminaries: VLAs with Conditional Flow Matching Action Head.** Given empirical observations $A^0 \sim \pi^0$ and $A^1 \sim \pi^1$, a flow defines an Ordinary Differential Equation (ODE) $dA^\tau = v(A^\tau, \tau)d\tau$ on $\tau \in [0, 1]$, with $v$ the velocity field to push $A^0$ to $A^1$, following the path defined by the ODE (Lipman et al., 2022). In VLA, $\tau = 0$ corresponds to a simple action distribution $A_t^0 = \epsilon \sim \mathcal{N}(0, \mathbf{I})$, and flows to an action from the dataset $A_t^1 \sim p(\cdot|\mathbf{o}_t), (A_t^1, \mathbf{o}_t) \in \mathcal{D}$ at $\tau = 1$. Assuming linear-Gaussian transformation, the action at time $\tau$ is formulated as $A_t^\tau = \mathcal{N}(\tau A_t^1, (1-\tau)\mathbf{I})$. To accelerate the learning process, rectified flow (Liu, 2022) assume linear ODEs for straight flow which is easier to model and compute. The action at time $\tau$ is then a linear interpolation $A_t^\tau = \tau A_t^1 + (1-\tau)A_t^0$. The goal is to learn a velocity field $v_\theta(A_t^\tau, \mathbf{o}_t)$ parameterized by $\theta$ to match the rectified flow $u(A_t^\tau|A_t) = A_t^0 - A_t^1 = \epsilon - A_t^1$. This yields the objective of $\pi_0$ (Black et al., 2024):

$$\min_\theta \mathcal{L}_{\pi_0}^\tau = \mathbb{E}_{p(A_t^1|\mathbf{o}_t), q(A_t^\tau|A_t^1)}||v_\theta(A_t^\tau, \mathbf{o}_t) - u(A_t^\tau|A_t^1)||^2. \tag{2}$$

**Worst-Case Action Noise.** We define worst-case action noise in VLAs as an $\ell_p$-bounded noise that maximally reduces the success rate. In robust RL literature, since success rate is hard to measure per step, researchers assume existing well-trained policy have high chance to succeed, and maximizing deviation to current policy leads to successful attacks (Huang et al., 2017). In VLAs, offline demonstrations $(\mathbf{o}_t, A_t)$ provide actions that are more likely to lead to success and can be learned through flow matching. Therefore, we use the loss of the flow matching objective, $||v_\theta(\hat{A}_t^\tau, \mathbf{o}_t) - u(\hat{A}_t^\tau|\hat{A}_t^1)||^2$, as an empirical measure of action quality. We also conduct a pilot study in Appendix. C.5, which empirically shows flow matching loss is strongly correlated with success rate, with significant Pearson correlation of $r = -0.95, p < 0.05$.

Formally, we define the perturbed action as $\hat{A}_t^1 = A_t^1 + \delta$, where a noise $\delta$ is added action noise. In this noisy flow, the perturbed action at time $\tau$ is $\hat{A}_t^\tau = \tau \hat{A}_t^\tau + (1-\tau)A_t^0$, and the perturbed rectified flow is $u(\hat{A}_t^\tau|\hat{A}_t^1) = u(A_t^\tau|A_t^1) - \delta$. Plugging this into the original $\pi_0$ objective, we got:

$$\delta \in \arg\max_\delta \mathbb{E}_{p(A_t^1|\mathbf{o}_t), q(\hat{A}_t^\tau|\hat{A}_t^1)}||v_\theta(\hat{A}_t^\tau, \mathbf{o}_t) - u(\hat{A}_t^\tau|\hat{A}_t^1)||^2$$
$$= \arg\max_\delta \mathbb{E}_{p(A_t^1|\mathbf{o}_t), q(\hat{A}_t^\tau|\hat{A}_t^1)}||v_\theta(\hat{A}_t^\tau, \mathbf{o}_t) - u(A_t^\tau|A_t^1) - \delta||^2. \tag{3}$$

Here, the noise on $v_\theta(\hat{A}_t^\tau, \mathbf{o}_t)$ perturbs the action input, enforcing Lipschitz continuity of $v_\theta$ to stabilize action outputs under perturbations. Noise on $u(\hat{A}_t^\tau | \hat{A}_t^1)$ maximizes loss in the direction $v_\theta(\hat{A}_t^\tau, \mathbf{o}_t) - u(A_t^\tau | A_t^1)$, where velocity field and rectified flow diverge the most. In practice, we compute $\delta$ via PGD (Madry et al., 2017), a gradient-based attack method.

**Robustness Against Action Noise.** With the worst-case action noise $\delta$, enhancing robustness against worst-case $\delta$ requires minimizing the conditional flow matching objective against the worst-case noise. To stabilize the result without noise, we use the TRADES objective that maintain the original $\pi_0$ loss, which provides optimal tradeoff between robustness and accuracy (Zhang et al., 2019):

$$\min_\theta \mathcal{L}_{\pi_0}^\tau + \mathcal{L}_{out}^\tau = \min_\theta \left[ \mathcal{L}_{\pi_0}^\tau + \lambda_{out} \max_\delta \mathbb{E}_{p(A_t^1|\mathbf{o}_t), q(\hat{A}_t^\tau|\hat{A}_t^1)} ||v_\theta(\hat{A}_t^\tau, \mathbf{o}_t) - u(\hat{A}_t^\tau|\hat{A}_t)||^2 \right], \quad (4)$$

where $\lambda_{out}$ is a hyperparameter to control the balance between flow matching without perturbation and the robustness against noisy VLA output.

**Remark 1.** Robustness against action noise can be understand as flow matching against both clean and noisy action distribution. The model continues to match the clean distribution sampled from $p(A_t|\mathbf{o}_t)$ while additionally accounting for an adversarially perturbed alternative $p(A_t + \delta|\mathbf{o}_t)$. Training with such perturbations prepares the model for test-time noise and improves robustness.

**Remark 2.** Robustness against action noise can also be interpreted as a form of label smoothing (Müller et al., 2019). Injecting noise into actions makes the learned flow less certain, discouraging overconfident decisions and overfitting to specific actions. This yields better generalization and encourages more stochastic behavior that covers a wider range of plausible actions, a property shown to support robust decision-making in practice (Eysenbach & Levine, 2021).

**Remark 3.** Robustness against action noise can be understand by penalizing outliers. Recall the noise $\delta$ approximately points in the direction $v_\theta(\hat{A}_t^\tau, \mathbf{o}_t) - u(A_t^\tau|A_t^1)$, where the velocity field and rectified flow mismatch the most. Consequently, any mismatch during training is amplified quadratically by $\delta$ in the MSE objective of flow matching. The objective therefore acts to penalize outliers that the VLA cannot fit well, improving performance on corner cases and reducing non-robust failure modes.

**Generalizing to Other VLAs.** For general diffusion-based VLAs, just ignore the assumption of $u(\hat{A}_t^\tau|\hat{A}_t^1) = u(A_t^\tau|A_t^1) - \delta$ of rectified flow and use the first line of Eqn. 3 to generate the worst-case action perturbation. For autoregressive VLAs, take OpenVLA as an example, we perturb the actions before binning to maximize its cross entropy loss, with robust objective following Eqn. 4. The perturbations are constrained so that the action output remains within the original bin and its adjacent bins. This ensures that, even when errors occur, the outputs stay in the neighborhood of the correct action and the worst-case robustness is improved.

## 4.2 ROBUSTNESS AGAINST VLA INPUTS

Next, we study robustness to noisy VLA inputs. These noises can have different types in varying modalities. We observe that these perturbations do not alter the underlying task semantics, so the optimal action should remain unchanged. We therefore encourage the flow-matching objective to produce similar actions under perturbed inputs. To automatically handle diverse observation types, we cast the perturbation selection as a multi-armed bandit problem and use an upper confidence bound (UCB) algorithm (Auer, 2002) to identify the most harmful noise for adversarial training.

**Robustness to Single Noise.** Given a VLA policy $\pi(A_t|\mathbf{o}_t)$, input variation can stem from various sources, including sensory noise and camera error in observation, or irrelevant objects, lighting variations in external environment. Although inputs may vary, the robot's underlying state does not change. That is, the robot operates in the same physical world to execute the same task. Consequently, the optimal actions should remain unchanged. For an input perturbation $\omega^i \in \Omega$, we define the flow-matching objective under each input noise as:

$$\min_\theta \max_{\omega^i} \mathbb{E}_{p(A_t|\mathbf{o}_t), q(A_t^\tau|A_t^1)} ||v_\theta(A_t^\tau, \omega^i(\mathbf{o}_t)) - u(A_t^\tau|A_t)||^2. \quad (5)$$

Here, we adversarially select $\omega^i$ to simulate worst-case inputs. If the input noise is fixed, we leave it unchanged. This objective is inspired by state-perturbation techniques in offline RL (Shen et al., 2020; Yang et al., 2022). Now, it remains unknown how to balance many perturbations automatically.

**Balancing Various Noises.** While RobustVLA targets robustness against a broad set of perturbations, it remains unclear *which* perturbation types contributes most to overall performance. For instance, simple Gaussian noise can be easy to defend, yet confer little robustness to more complex noises. Assigning fixed weights to each perturbation type is possible, but manually tuning the weight for each uncertainty can be time-consuming. We therefore seek to automatically maximize overall robustness by adaptively selecting the input perturbation at each training iteration.

This selection problem can be naturally formulated as a multi-armed bandit, with the goal of maximizing the overall robustness of the VLA policy. At each training step $n$, the algorithm selects an uncertainty $\omega^i \in \Omega$ to train the objective in Eq. 6. The UCB algorithm (Auer, 2002) provides a principled way to solve multi-armed bandits. Given the times each uncertainty has been explored as $\omega^i(n)$ at $n^{th}$ training iteration, the uncertainty is selected as:

$$\omega^i_* = UCB(\Omega, n) = \arg\max_{\omega^i \in \Omega} \left[ r_n(\omega^i) + \alpha \sqrt{\frac{\log(n)}{\omega^i(n)}} \right], \tag{6}$$

where $\alpha > 0$ is the exploration coefficient. To prioritize robustness, we define the reward as the increase in the flow-matching loss induced by the perturbation, *i.e.*, the gap between the noisy and clean objectives for the same $(\mathbf{o}_t, A_t)$ pair:

$$r_n(\omega^i) = \mathbb{E}_{p(A_t|\mathbf{o}_t), q(A_t^\tau|A_t^1)} ||v_\theta(A_t^\tau, \omega^i(\mathbf{o}_t)) - u(A_t^\tau|A_t)||^2 - ||v_\theta(A_t^\tau, \mathbf{o}_t) - u(A_t^\tau|A_t)||^2. \tag{7}$$

To stabilize the reward, we apply $z$-score normalization with the mean and standard deviation maintained by an exponential moving average, with a decay factor of 0.9. Equipped with the UCB-selected uncertainty type $\omega^i_*$, the training objective against diverse input noise becomes:

$$\min_\theta \mathcal{L}^\tau_{\pi_0} + \mathcal{L}^\tau_{in} = \min_\theta \mathcal{L}^\tau_{\pi_0} + \lambda_{in} \max_{\omega^i_*} \mathbb{E}_{p(A_t|\mathbf{o}_t), q(A_t^\tau|A_t^1)} ||v_\theta(A_t^\tau, \omega^i_*(\mathbf{o}_t)) - u(A_t^\tau|A_t)||^2, \tag{8}$$

where $\lambda_{in}$ balances the $\pi_0$ loss and the input robustness term. Finally, to enhance local smoothness of VLAs, we add a $\ell_p$ bounded observation noise $\eta$ to maximize the flow matching loss of $\mathcal{L}^\tau_{in}$ computed via PGD (Madry et al., 2017), which works well empirically.

**Generalizing to Other VLAs.** For other diffusion-based VLAs, our method can be applied by changing the rectified flow matching loss directly. For autoregressive VLAs, we apply the perturbation to observations and then map to action tokens, the remaining procedure is unchanged.

**Overall RobustVLA Loss.** Finally, we combine robustness to input and output perturbations with the $\pi_0$ objective. Let $\lambda_{in}$ and $\lambda_{out}$ be hyperparameters that weight the input- and output-robustness terms within $\mathcal{L}^\tau_{in}$ and $\mathcal{L}^\tau_{out}$, respectively. The overall training objective is:

$$\min_\theta \mathcal{L}^\tau_{RobustVLA} = \min_\theta \mathcal{L}^\tau_{\pi_0} + \mathcal{L}^\tau_{in} + \mathcal{L}^\tau_{out}. \tag{9}$$

The pseudocode of our RobustVLA is given in Appendix. D.

## 5 ROBUSTVLA EXPERIMENTS

**Experiment Setting.** Consistent with the robustness evaluation in Section 3.1, we assess all methods under the same 17 perturbations using the LIBERO benchmark and its recommended setup. We conduct our main experiment on $\pi_0$ backbone due to its superior robustness compared with OpenVLA. We compare BYOVLA (Hancock et al., 2025), an off-the-shelf robust VLA method designed for visual uncertainties as our baseline. Our approach is termed *RobustVLA*. For our main experiments, we also consider four ablations of our RobustVLA, including: (1) a domain randomization (DR) variant that trains $\pi_0$ using randomly selected VLA input perturbations, (2) our RobustVLA without input regularization (w/o in), (3) our RobustVLA without output regularization (w/o out), (4) our RobustVLA without UCB for balancing multiple input noises (w/o UCB). We also evaluate our RobustVLA on the OpenVLA backbone under the same setting. *All reported improvements are absolute gains in success rate, expressed in percentage points.*

**Implementation Details.** We follow the default training recipe for $\pi_0$ and OpenVLA as recommended in their original codebase. For our RobustVLA, we set $\lambda_{in}$, $\lambda_{out}$ as 1, action noise $\delta$ as 0.03, and observation noise $\eta$ as $8/255$. All baselines use the same set of hyperparameter and implementations. See implementation details and additional training details in Appendix B.

Table 1: Average success rate (%) under 17 noise types on LIBERO tasks, evaluated on $\pi_0$ backbone.

| Noise Modality | Noise type | $\pi_0$ | DR | BYOVLA | w/o in | w/o out | w/o UCB | RobustVLA(Ours) |
|---|---|---|---|---|---|---|---|---|
| w/o Noise | | 96.0 | 94.8 | 95.2 | 96.0 | 94.9 | 94.9 | 95.5 |
| Action | Uniform Noise | 63.5 | 61.2 | 62.0 | 67.3 | 66.8 | 69.5 | **69.8** |
| | Gaussian Noise | 31.4 | 30.1 | 32.1 | **36.3** | 31.9 | 35.9 | **36.0** |
| | Action Bias | 23.0 | 27.6 | 21.2 | **44.9** | 37.0 | 41.3 | **42.3** |
| | Random Flips | 52.7 | 48.5 | 51.6 | 56.9 | 54.0 | 57.6 | **58.7** |
| | Sudden Spikes | 51.7 | 54.2 | 51.1 | 58.5 | 52.7 | 62.3 | **59.7** |
| Observation | Gaussian Noise | 51.4 | 64.1 | 58.7 | 55.7 | **94.5** | 84.3 | **93.8** |
| | Dead Pixel | 20.8 | 54.5 | 43.1 | 40.8 | 90.8 | 78.5 | **93.8** |
| | Motion Blur | 93.7 | 88.7 | 95.2 | 94.1 | 95.0 | 95.2 | **95.5** |
| | Color Jitter | 61.7 | 54.0 | 54.2 | 53.9 | 58.7 | 62.3 | **69.5** |
| | Image Rotation | 73.3 | 85.9 | 77.7 | 64.1 | 94.0 | 70.7 | **94.4** |
| | Image Shift | 74.6 | 72.3 | 70.4 | 63.2 | 89.2 | 62.3 | **92.7** |
| Environment | External Force | 37.1 | 31.9 | 37.3 | 39.0 | 37.9 | 39.0 | **40.8** |
| | Irrelevant Objects | 93.1 | 80.8 | 93.7 | 91.2 | 89.9 | 91.2 | **94.2** |
| | Lighting Variation | 94.3 | 89.0 | 95.0 | 94.6 | 94.3 | 92.4 | **95.6** |
| Instruction | Lexical Transform | 78.7 | 78.8 | 79.5 | 81.5 | 77.7 | 77.1 | **91.3** |
| | Syntactic Transform | 84.7 | 72.3 | 85.8 | 84.0 | 82.3 | 83.6 | **93.9** |
| | Adversarial Prompts | 79.2 | 56.7 | 80.1 | 75.7 | 72.2 | 74.6 | **80.2** |
| Average | | 62.6 | 61.8 | 64.0 | 64.8 | 71.7 | 69.3 | **76.6** |

## 5.1 ROBUSTVLA ON $\pi_0$ BACKBONE

**RobustVLA is more robust on $\pi_0$ backbone.** We compare RobustVLA against all non-ablated baselines under 17 perturbations. As shown in Table. 1, RobustVLA gains higher robustness on all 17 perturbations, outperforming $\pi_0$ by 14.0% and baseline BYOVLA by 12.6% on average robustness. Under clean conditions without perturbation, RobustVLA remains competitive with $\pi_0$ (95.5% vs. 96.0%), showing robustness gains do not come at the expense of clean performance. Notably, our method also generalizes to unseen perturbations not encountered during training, such as external forces, demonstrating robustness beyond the set of uncertainties considered during training. A separate analysis on LIBERO-long highlights the potential of RobustVLA on complex and long-horizon tasks, outperforming $\pi_0$ by 19.61% in success rate. See results in Appendix. C.3. Results also confirmed action as the modality worth further investigation, with our method achieving +8.0% success rate on average, comparing with +17.2% increase in uncertainties applied to VLA input.

**Analysis on ablations.** We further compare RobustVLA with ablated variants that remove input regularization (*Ours w/o in*) or output regularization (*Ours w/o out*). RobustVLA achieves higher robustness than both ablations in 14/17 perturbations. This suggests **robustness aimed at one modality can also benefit others**. To explain, output robustness helps counter action drift induced by input noise, whereas input robustness improves generalization to the unseen transitions induced by action noise. As an evidence, even without input regularization, *Ours w/o in* retains notable gains on Dead Pixel, an observation perturbation. Similarly, removing output regularization, *Ours w/o out* still gains robustness on Uniform Noise and Action Bias, two perturbations on actions. These results implies multi-modal robustness as a promising route toward generally robust VLAs.

For DR and *Ours w/o UCB*, several patterns emerge. First, DR performs poorly. While DR improves robustness to some observation noises, it fails under most environment and instruction noises, likely because it overfits to a small subset of easy perturbations. Second, *Ours w/o UCB* shows a 7.3% drop in robustness, indicating that UCB is crucial for balancing multiple perturbations rather than letting the model overfit to a single dominant noise source. Comparing with *Ours w/o in* that trains on output noise only, adding exposure to diverse input noises adds further robustness gains, even without UCB.

## 5.2 DISCUSSIONS

In this section, we demonstrate that RobustVLA is consistently effective on the OpenVLA backbone, achieves greater computational efficiency than the visual-robust BYOVLA, and maintains robustness under mixed perturbations applied on input and output.

**RobustVLA is also robust on OpenVLA backbone.** Beyond $\pi_0$, RobustVLA is also consistently effective on the autoregressive-based OpenVLA. As shown in Fig. 4a, RobustVLA improves average robustness by 13.2% over OpenVLA, surpasses the baseline BYOVLA by 10.4%, demonstrating the effectiveness of RobustVLA across diffusion-based and autoregressive VLAs. The detailed numerical results are presented in Appendix C.1.

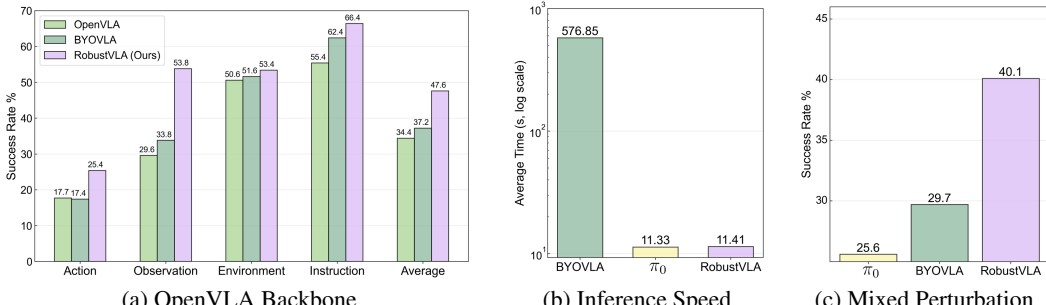

(a) OpenVLA Backbone     (b) Inference Speed     (c) Mixed Perturbation

Figure 4: RobustVLA improves robustness on the OpenVLA backbone, achieves fast inference speed, and withstands mixed perturbations.

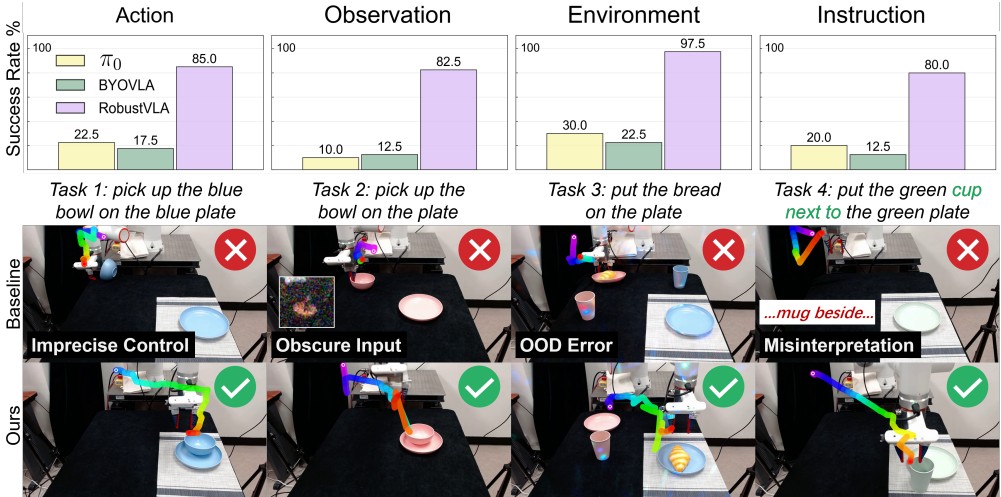

Figure 5: Real-world robustness results. Our RobustVLA is highly effective with scarce demonstrations, while baselines fail due to imprecise action control, obscure observation input, OOD observation and language misinterpretations.

**RobustVLA is computationally efficient.** As shown in Fig. 4b, RobustVLA achieves a per-episode inference time of $\sim 11$s, comparable to $\pi_0$ since they share the same architecture and parameter size. In contrast, RobustVLA is $50.6\times$ more efficient than BYOVLA, a visual-robust VLA method. This large gap arises because BYOVLA relies on a visual sensitivity probe, which requires multiple forward passes to identify sensitive visual regions of current VLAs and makes repeated calls to external LLMs to modify these regions, resulting in significant computational overhead.

**RobustVLA is robust against mixed perturbations applied on input and output.** To test this, we randomly sampled one uncertainty in VLA input and one uncertainty in VLA output, from categories defined in Fig. 2. We test the results on $\pi_0$ backbone, with random seed fixed to ensure all baselines were tested under the same set of uncertainties. As shown in Fig. 4, RobustVLA achieved a 14.5% higher robustness than $\pi_0$ and gains 10.4% robustness than baseline BYOVLA. These improvements are statistically significant ($p < 0.001$, paired-sample t-test).

## 5.3 REAL-WORLD EXPERIMENTS

**Real-world setup.** We deploy VLAs on a Fairino FR5 robotic arm with a Hitbot Z-EFG-C50 gripper. Visual inputs are provided by a ZED2 external camera and an Intel RealSense 435i wrist-mounted camera. We design four tasks to assess real-world performance: (1) pick up the blue bowl on the blue plate; (2) pick up the bowl on the plate with randomized colors; (3) place the bread on the plate; and (4) place the green cup next to the green plate. These tasks test VLA's capability on basic grasping, semantic generalization, deformable object manipulation, and spatial reasoning. For each task, we fine-tune $\pi_0$ with 25 demonstration trajectories. All methods achieve 100% success without perturbation after fine-tuning.

**Real-world noise.** We design physical noise sources for robustness evaluation. For environmental noise, we vary lighting by adding a neon bubble and introduce irrelevant objects into the testing scene. For action noise, we perturb motor calibration and introduce interference in serial communication. For instruction noise, we use a speech recognition model to process human speech containing dialects, unusual word order, or irrelevant content. For observation noise, we apply the same perturbations as in simulation. However, we omit *external forces* since they could physically damage the robot. For each noise modality, we conduct 10 trials per task and report aggregated results over 4 tasks.

**Real-world performance.** As shown in Fig. 5, RobustVLA is highly effective in real-world deployment, surpassing the best baseline by 65.6% in success rate. We attribute this to the limited 25 demonstrations of real-world trajectories available for fine-tuning. Unlike large and diverse simulation datasets such as LIBERO, laboratory-collected data are costly and offer limited state coverage, leading baseline VLAs to overfit demonstrations and fail under perturbations. In contrast, by anticipating noise during training, RobustVLA anticipates potential noise during training and remains robust to unexpected real-world disturbances. See per-task results in Appendix C.6.

**Failure Analysis.** Finally, we analyze the failure mode of baselines, aggregating $\pi_0$ and BYOVLA due to their similar behaviors. As illustrated in Fig. 5: (1) under *action* uncertainties, baselines exhibit imprecise control, resulting in the gripper unable to grasp the bowl precisely and occasionally knocking the bowl over; (2) under *observation* uncertainties, input noise obscures the visual signal, making the robot unable to identify the object clearly; (3) under *environment* shifts, the added neon light produces a severe illumination change that renders observations OOD; and (4) under *instruction* noise, speech parsing errors lead to misinterpretation, making robots placing the cup "beside" the table. In contrast, RobustVLA remains reliable in most trials, achieving consistent success in the real world. See demo videos in our codebase.

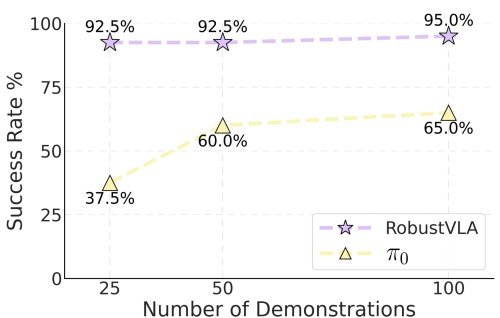

Figure 6: Results with more demonstrations. Although $\pi_0$ becomes more robust as additional data is provided, RobustVLA already achieves strong robustness in the low-data regime and consistently outperforms $\pi_0$ across all data scales.

**Results with More Demonstrations.** We evaluate our performance with 50 and 100 demonstrations in Task 1 to show results with sufficient data. While all methods achieves 100% success rate, as shown in Fig. 6, $\pi_0$ **gains robustness with additional data**. The success rate of $\pi_0$ increases from 37.5% at 25 demos to 60.0% at 50 demos and 65.0% at 100 demos. This shows real-world robustness benefits from additional trajectories with higher data diversity. Comparing with baselines, **our RobustVLA is highly robust in low-data regime, and consistently outperforms** $\pi_0$ **across all data scales.** Our success rate reaches 92.5% at 25 demos, 92.5% at 50 demos and 95.0% at 100 demos, while the baseline plateaus at 65.0%, achieving 30% robustness gain than $\pi_0$, showing RobustVLA provides substantial robustness gains beyond what is achievable through additional demonstrations alone.

## 6    CONCLUSION

In this paper, we evaluate and enhance the robustness of VLA models under multi-modal uncertainties. We evalute mainstream VLAs against 17 perturbations across four modalities, showing that actions are the most fragile, existing visual-robust VLAs fail to generalize beyond vision, and $\pi_0$ the most robust backbone. Building on these insights, we propose RobustVLA, a unified framework against input and output perturbations. For output robustness, we perform offline optimization against worst-case action noise that maximizes flow mismatch. For input robustness, we enforce consistent actions across semantically equivalent inputs and automatically identify the most harmful perturbation using UCB. On LIBERO, RobustVLA delivers absolute gains of 12.6% on $\pi_0$ and 10.4% on OpenVLA across 17 perturbations, achieves 50.6x faster inference than visual-robust BYOVLA, and improves robustness by 10.4% under mixed perturbations. Real-world FR5 robot under perturbation shows our RobustVLA is highly robust with limited data, outperforming $\pi_0$ by 65.6% success rate with 25 demos. With sufficient demos, our RobustVLA continues to maintain high robustness, achieving 30% higher success rate than $\pi_0$.

## 7 ACKNOWLEDGEMENT

This study was supported in part by the National Natural Science Foundation of China Project No. 62476018 and 62322318 and in part by the InnoHK initiative of the Innovation and Technology Commission of the Hong Kong Special Administrative Region Government via the Hong Kong Centre for Logistics Robotics, and in part by National Key R&D Program of China Project 2022ZD0161100.

## 8 ETHICS STATEMENT

Our research focuses on evaluating and enhancing the robustness of VLA models. By systematically studying their vulnerabilities under diverse uncertainties and proposing methods for robustness improvement, our work assists researchers and engineers in developing and deploying VLAs in environments where safety, stability, and reliability are critical. We emphasize that our study does not introduce new adversarial attack techniques, and is designed to provide constructive insights that strengthen the trustworthiness of embodied AI systems.

## 9 REPRODUCIBILITY STATEMENT

We have open sourced all our code for robustness evaluation and enhancement in `https://github.com/gakakulicc/RobustVLA`. Additional experiment details are available in Appendix A and B.

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

# APPENDIX FOR "ON ROBUSTNESS OF VISION-LANGUAGE-ACTION MODEL AGAINST MULTI-MODAL PERTURBATIONS"

**Declaration of LLM usage.** We use LLM to polish text only and authors have carefully checked all contents in the paper.

## A  EVALUATION UNCERTAINTIES

To systematically evaluate the robustness of VLAs, we consider uncertainties in actions, observations, environment, and instructions. For each type of uncertainties, we first relate it to practical sources of uncertainty in real-world, provide a formal definition, and describe the implementation details.

As defined in Section. 3, we define robustness of VLAs against perturbations $\omega \in \Omega$, where $\Omega = \{\Omega_\psi, \Omega_o, \Omega_a, \Omega_p\}$ corresponds to langauge, observation, action and environment dynamic, respectively. The perturbed variables are denoted as $\hat{\psi}$, $\hat{o}_t$, $\hat{A}_t$, and $\hat{\mathcal{P}}$.

### A.1  ACTION UNCERTAINTIES

In this paper, we consider 5 action noises related to sensorimotor noise, actuator wear and unexpected perturbations. A summary of these threats are available at Table. 2. Formally, we use $\mathbf{A}_t \in \mathbb{R}^d$ to denote the action vector produced by the policy at time step $t$, where $A_{t,i}$ is its $i$-th component. We use $\hat{\mathbf{A}}_t$ (with components $\hat{A}_{t,i}$) to denote the perturbed action after injecting noise. We describe details of each noise below.

Table 2: Summarization of action uncertainties.

| Uncertainty Type | Mathematical Formulation | Practical Sources |
|---|---|---|
| **Uniform Noise** | $\hat{A}_t = A_t + \epsilon, \quad \epsilon \sim \mathcal{U}(-\sigma, \sigma)^d$ | Sensorimotor Noise |
| **Gaussian Noise** | $\hat{A}_t = A_t + \epsilon, \quad \epsilon \sim \mathcal{N}(\mathbf{0}, \sigma^2 \mathbf{I})$ | Sensorimotor Noise |
| **Action Bias** | $\hat{A}_t = A_t + \sigma \cdot \mathbf{1}$ | Actuator Wear |
| **Random Flips** | $\hat{A}_{t,i} = \begin{cases} 1 & \xi_i < p \wedge \zeta_i < 0.5 \\ -1 & \xi_i < p \wedge \zeta_i \geq 0.5 \\ A_{t,i} & \text{otherwise} \end{cases}$ | Unexpected Perturbations |
| **Sudden Spikes** | $\hat{A}_{t,i} = \begin{cases} A_{t,i} + \sigma \cdot \text{sign}(\xi_i) & \|\xi_i\| < p \\ A_{t,i} & \text{otherwise} \end{cases}$ | Unexpected Perturbations |

**Uniform Noise.** Uniform noise is a random perturbation uniformly distributed within a fixed interval. It simulates sensorimotor noise, such as bias in motor response or random interference in sensor measurements. In the evaluation, the value of $\sigma$ was set to 0.04.

$$\hat{A}_t = A_t + \epsilon, \quad \epsilon \sim \mathcal{U}(-\sigma, \sigma)^d$$

**Gaussian Noise.** Gaussian noise is a random perturbation drawn from a normal distribution, with fluctuations centered around zero. It simulates sensorimotor noise such as sensor thermal noise, micro-vibrations, or irregular actuator responses. In the evaluation, the value of $\sigma$ was set to 0.3.

$$\hat{A}_t = A_t + \epsilon, \quad \epsilon \sim \mathcal{N}(\mathbf{0}, \sigma^2 \mathbf{I})$$

**Action Bias.** Action bias introduces a fixed offset across all action dimensions, resulting in consistent deviations from the intended control signal. It models actuator wear over time, or calibration drift and joint zero-point misalignment. In the evaluation, the value of $\sigma$ was set to 0.03.

$$\hat{A}_t = A_t + \sigma \cdot \mathbf{1}$$

**Random Flips.** Random flips replace selected action components with extreme values, creating abrupt deviations from normal behavior. This mimics unexpected disturbances such as sudden shocks from external disturbances, communication bit flips or actuator sticking and slipping. In the evaluation, the probability $p$ was set to 0.05.

$$\hat{A}_{t,i} = \begin{cases} 1 & \xi_i < p \wedge \zeta_i < 0.5 \\ -1 & \xi_i < p \wedge \zeta_i \geq 0.5 , \\ A_{t,i} & \text{otherwise} \end{cases} \quad \xi_i, \zeta_i \sim \mathcal{U}(0,1)$$

**Sudden Spikes.** Sudden spikes are abrupt, high-amplitude perturbations that occur with a certain probability. They simulate unexpected perturbations such as actuator jitter, control signal surges, or abrupt shocks in the mechanical system. In the evaluation, the probability $p$ was set to 0.05 and the spike magnitude $\sigma$ was set to 1.

$$\hat{A}_{t,i} = \begin{cases} A_{t,i} + \sigma \cdot \text{sign}(\xi_i) & |\xi_i| < p, \quad \xi_i \sim \mathcal{U}(0,1) \\ A_{t,i} & \text{otherwise} \end{cases}$$

## A.2 Observation Uncertainties

In this paper, we consider 6 observation noises related to sensory noise and camera error. A summary of these threats are available at Table. 3. Formally, we use $\mathbf{o}_t \in \mathbb{R}^{H \times W \times C}$ to denote the observation (image) received by the policy at time step $t$, where $o_{t,i,j}$ denotes the pixel intensity at location $(i,j)$. We use $\hat{\mathbf{o}}_t$ (with components $\hat{o}_{t,i,j}$) to denote the corrupted observation after injecting noise. We describe details of each noise below.

Table 3: Summarization of Observation Uncertainties.

| Uncertainty Type | Mathematical Formulation | Practical Sources |
|---|---|---|
| **Gaussian Noise** | $\hat{o}_t = \text{clip}(o_t + \epsilon, 0, 255), \quad \epsilon \sim \mathcal{N}(\mathbf{0}, \sigma^2 \mathbf{I})$ | Sensory Noise |
| **Dead Pixel** | $\hat{o}_{t,i,j} = \begin{cases} 255 & \xi_{i,j} < p \wedge \zeta_{i,j} < 0.5 \\ 0 & \xi_{i,j} < p \wedge \zeta_{i,j} \geq 0.5 \\ o_{t,i,j} & \text{otherwise} \end{cases}$ | Sensory Noise |
| **Motion Blur** | $\hat{\mathbf{o}}_t = \mathbf{o}_t * G_\sigma$ | Camera Error |
| **Color Jitter** | $\hat{\mathbf{o}}_t = \mathcal{S}_\alpha \circ \mathcal{C}_\beta \circ \mathcal{B}_\delta(\mathbf{o}_t)$ | Sensory Noise |
| **Image Rotation** | $\hat{\mathbf{o}}_t[i,j] = \mathbf{o}_t[R_\theta^{-1}(i,j)]$ | Camera Error |
| **Image Shift** | $\hat{\mathbf{o}}_t[i,j] = \mathbf{o}_t[i - \Delta i, j - \Delta j]$ | Camera Error |

**Gaussian Noise.** Gaussian noise is an additive perturbation where pixel values fluctuate around the mean according to a normal distribution. It simulates sensory noise including thermal fluctuation, dark current, and stochastic variations in imaging sensors. In the evaluation, the noise standard deviation $\sigma$ was set to 70.

$$\hat{\mathbf{o}}_t = \text{clip}(\mathbf{o}_t + \epsilon, 0, 255), \quad \epsilon \sim \mathcal{N}(\mathbf{0}, \sigma^2 \mathbf{I}_{HWC})$$

**Dead Pixel.** Dead pixel noise is an impulsive perturbation that forces certain pixels to extreme values (0 or 255). It simulates sensory errors such as dead pixels, stuck pixels, or bit errors in image transmission. In the evaluation, the corruption probability $p$ was set to 0.1.

$$\hat{o}_{t,i,j} = \begin{cases} 255 & \xi_{i,j} < p \wedge \zeta_{i,j} < 0.5 \\ 0 & \xi_{i,j} < p \wedge \zeta_{i,j} \geq 0.5 , \quad \xi_{i,j}, \zeta_{i,j} \sim \mathcal{U}(0,1) \\ o_{t,i,j} & \text{otherwise} \end{cases}$$

**Motion Blur.** Motion blur is a smoothing perturbation introduced by spatial convolution with a Gaussian kernel. It simulates camera error include shake, defocus, or object motion relative to the

camera. We denote the blur standard deviation by $\sigma$, the convolution kernel size by $K$, and the kernel half-width by $k = \frac{K-1}{2}$, so that the kernel spans indices $u, v \in [-k, k]$. In our evaluation, we set $\sigma = 1$ and $K = 5$.

$$\hat{\mathbf{o}}_t[i, j] = \sum_{u=-k}^{k} \sum_{v=-k}^{k} G_\sigma(u, v)\, \mathbf{o}_t[i - u, j - v], \quad G_\sigma(u, v) = \frac{1}{2\pi\sigma^2} e^{-\frac{u^2 + v^2}{2\sigma^2}}$$

**Color jitter.** Color jitter is a composite perturbation that adjusts image brightness, contrast, saturation, and sharpness. It simulates sensory errors like uneven gain in CMOS/CCD, lighting variation, white balance errors or automatic camera gain fluctuations. In the evaluation, the maximum perturbation factor was set to $0.4$ for all adjustments.

$$\hat{\mathbf{o}}_t = \mathcal{S}_\alpha \circ \mathcal{C}_\beta \circ \mathcal{B}_\delta(\mathbf{o}_t)$$

where $\mathcal{B}_\delta, \mathcal{C}_\beta, \mathcal{S}_\alpha$ represent the brightness, saturation, and sharpness enhancement functions, respectively. Their intensity parameters $\delta, \beta, \alpha$ are independent random perturbation factors sampled based on max_factor $= 0.4$.

**Image rotation.** Image rotation is a geometric perturbation that rotates pixels around the image center. It simulates camera error induced by robot tilting, changes in camera orientation, or misalignment in the mounting system. In the evaluation, the rotation angle $\theta$ was randomly sampled within a bounded interval $[-\sigma_\theta, \sigma_\theta]$. In the evaluation, the value of $\sigma_\theta$ was set to $20°$.

$$\hat{\mathbf{o}}_t[i, j] = \mathbf{o}_t[R_\theta^{-1}(i, j)], \quad R_\theta = \begin{bmatrix} \cos\theta & -\sin\theta \\ \sin\theta & \cos\theta \end{bmatrix}, \quad \theta \sim \mathcal{U}(-\sigma_\theta, \sigma_\theta)$$

**Image shift.** Image shift is a translation perturbation that displaces pixel coordinates by offsets proportional to the image dimensions. It simulates camera vibrations, miscalibration, or abrupt movements during perception. We denote the maximum shift fraction of the image dimensions by $\Delta_{\text{shift}}$. In the evaluation, the value of $\Delta_{\text{shift}}$ was set to $0.15$.

$$\hat{\mathbf{o}}_t[i, j] = \mathbf{o}_t[i - \Delta i, j - \Delta j], \quad \begin{aligned} \Delta i &\sim \mathcal{U}\big(-\Delta_{\text{shift}} H, \Delta_{\text{shift}} H\big) \\ \Delta j &\sim \mathcal{U}\big(-\Delta_{\text{shift}} W, \Delta_{\text{shift}} W\big) \end{aligned}$$

### A.3 Environmental Uncertainties

Environment uncertainties consider external influences. This can happen either in VLA input and output. In this paper, we consider external force in VLA output. For VLA input, we consider addition of irrelevant objects and lighting variations. This results in 3 environment uncertainties in total.

**External Force.** External force represents an exogenous disturbance applied directly to the robot's body or joints, rather than an error in its internal control signals. Unlike *action uncertainties*, which arise from sensorimotor noise or actuator imperfections affecting the executed commands, external forces originate from the environment and perturb the robot independently of its control policy. Such disturbances occur in real-world settings when a human pushes the robot, when the robot collides with obstacles, or when it interacts with moving objects in a cluttered environment. These forces are inherently non-deterministic in both timing and magnitude, making them a critical source of uncertainty during deployment. In the evaluation, we apply an external force $\mathbf{F}_{\text{external}}$ in addition to the control force $\mathbf{F}_{\text{control}}$:

$$\mathbf{F}_{\text{total}} = \mathbf{F}_{\text{control}} + \mathbf{F}_{\text{external}} \quad \text{where} \quad \mathbf{F}_{\text{external}}(t) = \begin{cases} \mathbf{F}_0 \cdot \mathbf{d} & t \in [t_i, t_i + \Delta t_i] \\ \mathbf{0} & \text{otherwise} \end{cases} \tag{10}$$

Here, $\mathbf{F}_0$ denotes the disturbance magnitude and $\mathbf{d}$ the direction vector. We set $\mathbf{F}_0 = 200$ N along $(1, 0, 0)$ (x-axis), with each application lasting $5 \pm 2$ timesteps and occurring at random intervals of 40–50 steps to emulate unpredictable external perturbations.

**Irrelevant Objects.** To evaluate the robustness of VLA models, we introduced additional distractor objects into the environment. Specifically, during task execution we placed assets drawn from unrelated tasks in close proximity to either the target object or the designated goal location. The number of distractors was fixed to three. We note that, in our experiments, this setting already

approaches the upper bound supported by the LIBERO environment: adding more distractors frequently leads to spatial conflicts during initialization.

**Lighting Variations.** Lighting variation is an observation-level uncertainty caused by fluctuations in natural or artificial illumination. Such changes may occur in real-world settings due to moving light sources, shifting daylight, or shadows cast by dynamic objects in the environment, all of which can significantly alter the visual appearance of a scene. To simulate these effects, we adopt the Phong reflection model, where the total intensity at a surface point is given by:

$$I = I_a + I_d + I_s, \tag{11}$$

$$I_d = k_d \cdot I_{\text{light}} \cdot \max(0, \mathbf{n} \cdot \mathbf{l}), \tag{12}$$

with $I_a$, $I_d$, and $I_s$ denoting the ambient, diffuse, and specular components, $k_d$ the diffuse reflection coefficient, $\mathbf{n}$ the surface normal, and $\mathbf{l}$ the direction vector to the light source.

In our evaluation, the illumination intensity $I_{\text{light}}$ was sampled from a Gamma distribution $\text{Gamma}(k, \theta)$ with parameters $k = 1.0$ and $\sigma^2 = 1.0$, producing both subtle and dramatic variations. To further emulate dynamic conditions, the light direction was updated every 3 simulation steps, with the azimuth angle $\theta \sim \mathcal{U}(0, 2\pi)$ while fixing the elevation at $45°$.

### A.4 INSTRUCTION UNCERTAINTIES

In this paper, we consider 3 uncertainties, including word-level lexical transform, sentence-level syntactic transform and adversarial prompt with ambiguous or irrelevant transformations. Examples of our added uncertainties are available in Table. 4. We describe details of each noise below.

Table 4: Examples of instruction uncertainties transformations compared to the original instruction.

| Type | Instruction |
|---|---|
| Original | pick up the black bowl between the plate and the ramekin and place it on the plate |
| Lexical Transform | Retrieve the ebony bowl situated between the dish and the ramekin and deposit it onto the dish |
| Syntactic Transform | Could you pick up the black bowl that is between the plate and the ramekin, and then place it on the plate with care? |
| Adversarial Prompt | I think you can do IT, maybe? pick up the black bBBowl, between the plate and the ramekin and place it on the plate |

**Lexical Transform** simulates real-world variations in word usage, encompassing phenomena such as dialectal differences and synonym substitutions. This dimension introduces surface-level perturbations to characters or words while preserving core semantics and syntactic integrity. These transformations probe model resilience against lexical noise encountered in everyday communication.

**Syntactic Transform** addresses structural flexibility inherent in human language expression. It modifies phrase ordering, sentence patterns, and grammatical constructions, clause insertion, or punctuation alterations—without altering propositional meaning. This dimension tests model robustness against grammatical reconfigurations that retain identical semantic content.

**Adversarial Prompts** evaluates model sensitivity to contextual noise and communicative distractions. It introduces semantically irrelevant content (e.g., social media tags, extraneous clauses), accidental error(OCR misrecognitions, keyboard typos) , or adversarial manipulations (sentiment polarity flips) while maintaining surface fluency. This dimension mimics real-world scenarios where core information must be discerned amidst misleading signals.

# B  TRAINING SETTING

## B.1  IMPLEMENTATION DETAILS

This appendix details the implementation details used throughout our experiments. These settings were found to be effective and robust in our empirical evaluations. We provide them here to facilitate reproducibility and to serve as a reference for future work.

### B.1.1  BASE TRAINING PARAMETERS

The foundational training hyperparameters for our models are summarized in Table 5. For training RobustVLA on the $\pi_0$ benchmark, we maintained consistency with the original $\pi_0$ setup in terms of batch size and total training steps. To optimize computational efficiency, we employed a hybrid training strategy: the action expert (a 300M parameter Gemma model) was fully fine-tuned, while the Vision-Language Model (VLM) component was trained using Low-Rank Adaptation (LoRA). A similar LoRA-based approach was adopted for training OpenVLA models but with minor modifications to balance GPU memory usage and training efficiency.

Table 5: Base training hyper-parameters.

| Parameter | RobustVLA on $\pi_0$ | On OpenVLA |
|---|---|---|
| Batch Size | 32 | 16 |
| Training Steps | 30,000 | 30,000 |
| Action Expert Tuning | Full Fine-tune | - |
| VLM Tuning | LoRA | LoRA |

### B.1.2  UCB EXPLORATION PARAMETERS

The parameters for the Upper Confidence Bound (UCB) exploration strategy are listed in Table 6. The UCB algorithm encourages the agent to explore less-visited states by adding an exploration bonus to the value estimate. This bonus is inversely proportional to the visit count, promoting a balance between exploiting known rewarding paths and exploring new ones.

**ucb_exploration_coeff**: This coefficient controls the weight of the exploration bonus in the UCB calculation. A higher value encourages more exploration. We set it to 1.0 as a standard baseline.

**ucb_window_size**: This defines the size of the sliding window used to calculate recent visit counts. A finite window size allows the agent to "forget" old visits and re-explore states that haven't been visited recently, which is crucial in non-stationary environments.

**ucb_ema_decay**: This parameter sets the decay rate for an Exponential Moving Average (EMA) used to smooth the visit counts. A decay of 0.9 places more weight on recent visits, making the exploration bonus more adaptive to recent policy changes.

**ucb_min_samples**: The minimum number of samples required for a state before the UCB bonus is applied. This prevents underexplored states with very few samples from having an excessively high and uncertain bonus.

It is important to note that while these specific values were used effectively in our experiments, they were not extensively optimized. The UCB framework is highly extensible and possesses significant potential for task-specific optimization. Beyond the commonly used enhancement mechanisms, the UCB components can be easily augmented or pruned to suit particular tasks or application scenarios.

Table 6: UCB exploration hyperparameters.

| Parameter | Value |
|---|---|
| ucb_exploration_coeff | 1.0 |
| ucb_window_size | 100 |
| ucb_ema_decay | 0.9 |
| ucb_min_samples | 10 |

### B.1.3 Adversarial Training Parameters

The parameters for adversarial training, which includes both action-space and observation-space (image) perturbations, are provided in Table 7.

**adv_epsilon** ($\epsilon$): This is the maximum allowed perturbation magnitude, defining the $\ell_\infty$ norm ball around the original input (action or image) within which the adversarial example must lie. A larger $\epsilon$ creates a stronger but potentially less stealthy attack.

**pgd_steps**: The number of iterative steps used to generate the PGD attack. More steps typically lead to a more powerful adversarial example within the given $\epsilon$ constraint, as the attack can better orient itself towards the steepest ascent of the loss function.

**pgd_alpha**: The step size for each iteration of the PGD attack. It determines how much the perturbation is updated in each step. It is typically a fraction of $\epsilon$.

**loss_weight** ($\lambda$): This is the weight coefficient used to balance the input and output losses, which determines which measurement our optimization focuses more on for robustness.

While parameters below are generally not highly sensitive in our preliminary study, we offer the following guidance: the perturbation limits defined by `adv_epsilon_action` and `adv_epsilon_image` should not be set excessively large. We observed that even small adversarial perturbations during training are sufficient to confer strong robustness against larger noises during evaluation. To enhance the effectiveness of adversarial training, increasing the number of PGD steps (`pgd_steps_*`) is a recommended strategy. However, this improvement comes at the cost of increased GPU memory consumption and longer training times.

Table 7: Adversarial training hyperparameters.

| Parameter | Action Space | Observation Space |
|---|---|---|
| adv_epsilon | 0.03 | 8/255 |
| pgd_steps | 3 | 3 |
| pgd_alpha | 0.01 | 2/255 |
| $\lambda$ | 1 | 1 |

### B.2 Discussion on Parameter Selection

Table 8: Average success rate (%) of the ablation study on the PGD action space $\epsilon$ parameter.

| | $\pi_0$ | $\epsilon = 0.015$ | $\epsilon = 0.03$ **(used)** | $\epsilon = 0.06$ |
|---|---|---|---|---|
| Clean | 96.0 | 95.8 | 95.5 | 93.9 |
| Uniform Noise | 63.5 | 66.0 | **69.8** | 68.1 |
| Gaussian Noise | 31.4 | **37.1** | 36.0 | 35.9 |
| Action Bias | 23.0 | 38.5 | 42.3 | **43.7** |
| Random Flips | 52.7 | 56.5 | **58.7** | 56.8 |
| Sudden Spikes | 51.7 | 61.1 | 59.7 | **61.5** |
| **Average** | 44.5 | 51.8 | 53.3 | 53.2 |

Table 9: Average success rate (%) of the ablation study on the UCB parameters.

| | Ours | exp_coef=1.5 | ema=0.6 | win=200 |
|---|---|---|---|---|
| Clean | 95.5 | 93.8 | 95.5 | 96.2 |
| Gaussian Noise | 93.8 | 93.9 | 93.6 | 93.7 |
| Dead Pixel | 93.8 | 94.0 | 93.4 | 93.6 |
| Motion Blur | 95.5 | 95.6 | 95.2 | 95.3 |
| Color Jitter | 69.5 | 72.2 | 68.0 | 70.7 |
| Image Rotation | 94.4 | 95.0 | 93.8 | 94.6 |
| Image Shift | 92.7 | 93.5 | 92.0 | 93.0 |
| **Average** | 90.0 | 90.7 | 89.3 | 90.2 |

Our method involves two groups of hyperparameters: (1) the PGD-based adversarial action perturbation parameters, and (2) the UCB-based adaptive noise selection parameters. As stated in Appendix. B.1, the overall performance of RobustVLA is not highly sensitive to these hyperparameters. Nevertheless, we provide a more detailed discussion on initialization and tuning principles.

### B.2.1 PGD PARAMETERS

The parameters `pgd_alpha` and `pgd_steps` directly control the optimization strength of adversarial action search. In general, larger `pgd_steps` allow the perturbation to approach a stronger worst-case solution. However, this improvement comes with increased computational cost, and we recommend practitioners balance effectiveness and resource constraints—using as many steps as feasible within their budget.

The perturbation magnitude `adv_epsilon` is less intuitive to reason about, and its sensitivity requires empirical validation. Table 8 compares multiple values of $\epsilon$ and shows that the success rates remain stable across a broad range of values; all choices outperform the non-adversarial baseline by a significant margin. This confirms that `adv_epsilon` is not sensitive in practice. A value around 0.03 consistently provides strong robustness while avoiding over-perturbation, and we use it as the default.

Before introducing tuning guidelines, we provision the additional ablation results in Table 9. The UCB module shows strong robustness across different hyperparameter choices: varying the `ucb_exploration_coeff`, `ucb_ema_decay`, or `ucb_window_size` only leads to minor fluctuations in success rate. Larger `ucb_exploration_coeff` slightly improve performance on harder corruptions, while smaller `ucb_ema_decay` values make the selection more reactive but somewhat less stable. Enlarging the `ucb_min_samples` provides mild stability gains. These results confirm that our default configuration offers a well-balanced trade-off without requiring precise tuning.

For the UCB-based adaptive perturbation selection, a good starting point is the configuration reported in Appendix. B.1, which has demonstrated stable performance across diverse tasks and typical noise levels.

During tuning, we recommend a data-driven, stability-first strategy. If one observes high-frequency oscillation in the training loss or overly frequent switching in selected perturbations, it is advisable to increase stability by lowering `ucb_exploration_coeff`, lowering `ucb_ema_decay`, or enlarging `ucb_window_size`. Conversely, if the loss plateaus for an extended period or the selected perturbations become overly concentrated—indicating insufficient adaptation—these parameters can be adjusted in the opposite direction to encourage exploration. This closed-loop adjustment ensures a balance between adaptability and convergence stability.

### B.3 TRAINING LOSS

Figures. 7 and 8 report the training loss curves of $\pi_0$ under our hyperparameter configuration, using both LIBERO simulation data and real-world trajectories. Across all loss components—including the clean loss, input-consistency loss, and adversarial output loss—the curves remain smooth and stable throughout training. This indicates that the chosen PGD parameters and UCB-based perturbation scheduling provide a reliable optimization landscape without inducing instability or oscillatory behavior.

Given this stability, we consider the presented hyperparameter setting to be a robust default for training RobustVLA. If further tuning is required for new tasks or environments, practitioners can use these loss patterns as a reference:

**Excessive fluctuations or abrupt jumps** in any loss component typically suggest the need to reduce the UCB exploration coefficient, increase the window size, or lower the PGD step size.

**Prolonged plateaus** may indicate insufficient exploration or under-strength adversarial perturbation, in which case moderate increases to these parameters can help.

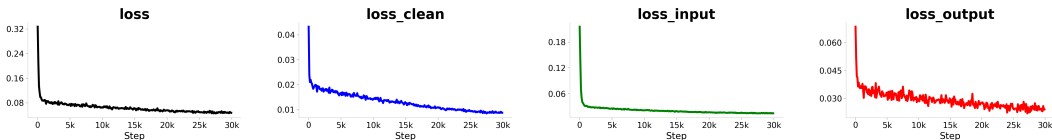

Figure 7: Loss during training with LIBERO data

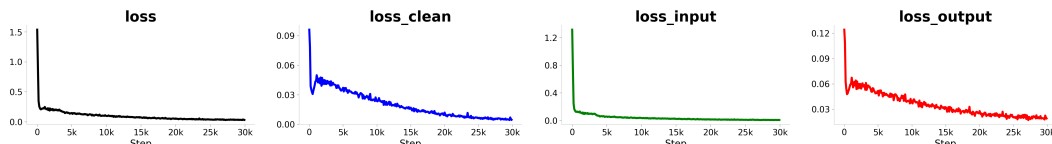

Figure 8: Loss during training with real-world data

Table 10: Average success rate (%) under 17 noise types on LIBERO tasks, evaluated on OpenVLA backbone.

| Noise Modality | Noise type | OpenVLA | BYOVLA | RobustVLA |
|---|---|---|---|---|
| Action | Uniform Noise | 25.4 | 24.2 | **37.6** |
| | Action Gaussian | 7.4 | 8.3 | **10.1** |
| | Action Bias | 11.8 | 12.6 | **24.9** |
| | Random Flips | 21.6 | 20.3 | **25.4** |
| | Sudden Spikes | 22.2 | 21.5 | **28.8** |
| Observation | Visual Gaussian | 0.8 | 1.5 | **60.9** |
| | Dead Pixel | 21.6 | 25.1 | **68.9** |
| | Color Jitter | 31.0 | 37.3 | **38.1** |
| | Image Rotation | 22.3 | 26.3 | **26.6** |
| | Image Shift | 42.9 | 46.6 | **47.3** |
| | Motion Blur | 59.3 | 66.1 | **80.9** |
| Instruction | Lexical Transform | 57.7 | 55.2 | **58.7** |
| | Syntactic Transform | 59.1 | 69.5 | **76.1** |
| | Adversarial Prompts | 49.3 | 62.6 | **64.5** |
| Environment | Irrelevant Objects | 72.3 | 72.7 | **77.0** |
| | Lighting Variation | 64.4 | **67.4** | 64.9 |
| | External Force | 15.0 | 14.6 | **18.2** |
| Average | | 34.4 | 37.2 | **47.6** |

## C DETAILS OF EXPERIMENTAL RESULTS

Due to space constraints in the main text, which focused primarily on experimental results and model performance across various robustness categories, more detailed data and secondary observations could not be included. Therefore, we provide in this appendix complementary details of our experimental outcomes to facilitate a more comprehensive understanding of this work.

### C.1 DETAILS OF OPENVLA EXPERIMENTAL RESULTS

We conducted comprehensive tests on OpenVLA, BYOVLA, and our enhanced RobustVLA models across our benchmark tasks. While the results analyzed by type of uncertainty have been discussed in detail in the main text, we provide the complete data details of our tests in Table 10.

### C.2 DETAILS AND DISCUSSION ON ROBUSTNESS TESTING OF $\pi_0$ AND $\pi_0$-FAST

In addition to the Fig. 3c presented in the main paper, we further compare $\pi_0$ and $\pi_0$-FAST under progressively increasing perturbation strengths. Although these two models share the same pre-trained VLM backbone and a similar policy architecture, they differ fundamentally in their action-generation mechanisms: $\pi_0$ employs a diffusion-based action expert, whereas $\pi_0$-FAST produces actions autoregressively via token prediction.

To examine whether this distinction leads to different robustness characteristics, we conduct stress tests on the two most fragile modalities identified in Fig. 3a: action perturbations and observation perturbations. For each modality, we gradually increase the perturbation magnitude and record the resulting drop in success rate.

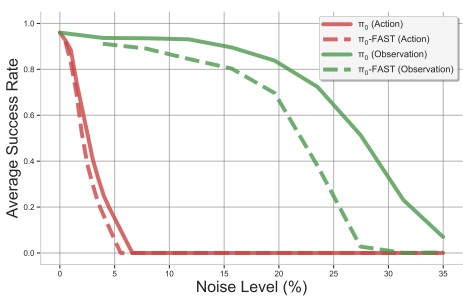

Figure 9: $\pi_0$ and $\pi_0$-FAST under varying noise level

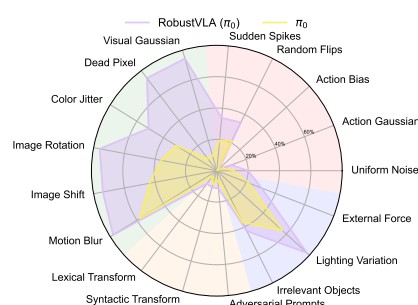

Figure 10: $\pi_0$ vs RobustVLA in LIBERO-long

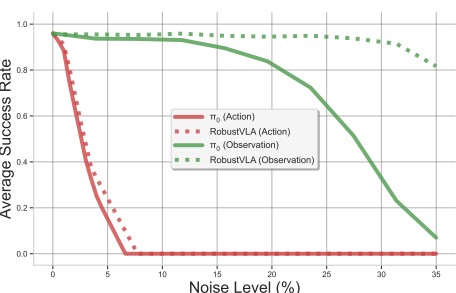

Figure 11: $\pi_0$ and RobustVLA under varying noise level.

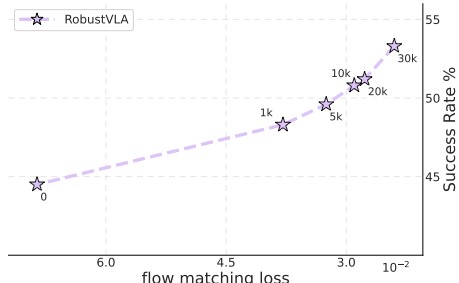

Figure 12: Effect of Flow Matching Loss on robustness of action modality.

As shown in Fig. 9, the diffusion-based $\pi_0$ consistently outperforms the autoregressive $\pi_0$-FAST across nearly all noise levels. The performance gap widens as perturbations become more severe, reinforcing our main-paper conclusion regarding robustness differences across VLA model classes. These results provide additional evidence that diffusion-based policies possess stronger intrinsic robustness to multi-modal disturbances than their autoregressive counterparts.

## C.3 DETAILS OF LONG-HORIZON TASK RESULTS

When evaluating complex and long-horizon tasks, LIBERO-long in the LIBERO suite as an appropriate benchmark. Therefore, we conduct an additional analysis in this section. As shown in Fig. 10, we observe that the inherently fragile action modality of the VLA $\pi_0$ deteriorates further on long-horizon tasks, while the instruction modality experiences a substantial drop due to compounding instruction-following errors. The robustness of the observation and environment modalities also declines to varying degrees. In contrast, RobustVLA maintains strong gains in the observation and environment modalities (39.93% and 12% on average, respectively), and further improves robustness in the more vulnerable action and instruction modalities (8% and 4.33% on average, respectively). Overall, RobustVLA achieves an average improvement of 19.61% across all four modalities on LIBERO-long.

## C.4 PERFORMANCE OF ROBUSTVLA UNDER VARYING NOISE LEVELS.

In the main text, we reported robustness improvements across the full set of uncertainty types and modalities. In this appendix subsection, we further examine how RobustVLA behaves within a single modality under varying noise intensities. We focus on the two most fragile modalities identified in our evaluation, action and observation, and systematically sweep their corresponding noise levels. As shown in Figure 11, RobustVLA consistently achieves positive robustness gains across all tested noise levels, with average improvements of 5.6% in the action modality and 23% in the observation modality. Moreover, we observe a clear trend in which robustness gains increase with noise intensity. For example, in the action modality, the improvement is only 1.3% at a noise level of 0.5% but rises to 8.7% at 5%. A similar pattern appears in observation perturbations, where the gain grows from

1.9% at a noise level of 3.9% to 74.7% at 35%. These results suggest that RobustVLA not only stabilizes performance under mild disturbances but also provides increasing benefits as perturbations become more severe.

## C.5 Correlation Between Adversarial Flow-Matching Loss and Action Robustness

To assess whether flow-matching loss serves as a meaningful proxy for action robustness, we analyze the correlation between our adversarial flow-matching loss and task success under action perturbations. Conceptually, flow-matching provides a differentiable surrogate for action quality, consistent with prior work in adversarial robustness where differentiable losses are used as proxies for task-level objectives that are not directly differentiable.

In VLA imitation-learning settings, credit assignment is particularly challenging due to high-frequency control. As a result, flow-matching naturally quantifies the deviation between the model's predicted actions and expert demonstrations. Larger flow-matching deviations correspond to larger mismatches from the demonstrated trajectories, which intuitively reduce the likelihood of successful task execution.

Following this motivation, we measure the adversarial flow-matching loss defined in Eq. 4 and compare it with the corresponding success rates under action perturbations. As shown in Fig. 12, the two quantities exhibit a clear monotonic trend. Quantitatively, the Pearson correlation coefficient is $r = -0.953$ with $p < 0.05$, indicating strong and statistically significant negative correlation. Intuitively, as adversarial flow-matching loss decreases ($0.0686 \rightarrow 0.0240$), the success rate increases correspondingly ($44.5\% \rightarrow 53.3\%$).

These results confirm that adversarial flow-matching loss is a reliable and interpretable indicator of action robustness: larger worst-case deviations in the action space directly translate into lower task success rates.

## C.6 Details of Real-World Experimental Results

We present in Fig. 13 the specific task success rates of various methods under different uncertainties for different tasks. Furthermore, by analyzing the robot's performance, we have preliminarily identified the causes of failures in baseline models under different robustness conditions, as well as the advantages exhibited by our method:

**Action Uncertainty.** The baseline models exhibit pronounced sensitivity to action noise, where even minor deviations—such as motor calibration errors, communication jitters, or actuator delays—cause large downstream failures. In practice, once a single action deviates from the intended trajectory, the autoregressive or imitation-trained policy is unable to recover, as the SFT data seldom contain recovery demonstrations or state-reset behaviors. This results in misaligned grasps, gripper jitter, or cumulative drift of end-effector poses. As shown in Fig. 5, the baseline often displays large-amplitude oscillations and severe localization offsets.

In contrast, RobustVLA explicitly optimizes output robustness (Eq. 4) by injecting worst-case action perturbations into the flow-matching objective during training. This exposes the model to both clean and noisy action distributions, effectively smoothing the action manifold and preventing brittle, single-point failures. As a result, RobustVLA produces significantly more stable controls and suppresses error accumulation from individual noisy actions.

**Observation Uncertainty.** Under visual corruptions such as Gaussian noise, dead pixels, motion blur, or illumination changes, baseline models frequently misinterpret object boundaries or execution states. For example, in tasks like "place the bread on the plate," motion blur can obscure the bread–plate contact, causing the baseline to repeatedly attempt a grasp even after the object has been placed. Although BYOVLA reduces distraction from irrelevant regions, its robustness is largely confined to static visual artifacts and remains insufficient for temporal or semantic consistency.

RobustVLA mitigates these issues through input-robustness regularization (Eq. 9), which enforces consistent actions under semantically preserving perturbations. Combined with our UCB-based adaptive perturbation scheduling, the model is trained against the most harmful visual corruptions,

enabling effective feature denoising and context-aware reasoning. Consequently, RobustVLA reliably infers state transitions and target positions even under severely degraded observations.

**Environment Uncertainty.** Environmental variations such as distractor objects, background changes, or dynamic illumination induce OOD shifts that significantly degrade baseline performance. Because imitation-based policies tend to overfit the limited demonstration distributions, they often mis-track objects, attend to irrelevant items, or misjudge their own pose when confronted with unseen layouts. In real-world tests, such failures occasionally culminate in catastrophic behaviors (e.g., descending directly toward the table when shadows alter perceived depth).

Our framework alleviates this failure mode by prioritizing perturbations that maximally increase the flow-matching loss, ensuring that the model repeatedly encounters difficult OOD scenarios during training. Exposure to these diverse perturbations improves target tracking and state estimation under environmental variations, resulting in robust execution even when scene layouts deviate significantly from training conditions.

**Instruction Uncertainty.** Baseline models are highly sensitive to linguistic variations such as synonym substitutions, word-order changes, or extraneous modifiers. Since these models often rely on shallow keyword matching rather than semantic grounding, injected linguistic noise frequently alters their interpretation of spatial relations or object references. For instance, substituting "put the cup next to the plate" with "gently place the cup by the side of the dish" causes the baseline to misidentify the target position or confuse object roles, leading to failed execution.

RobustVLA addresses this issue by enforcing cross-perturbation action consistency for language inputs as well. By treating semantically equivalent linguistic variants as invariances during training, RobustVLA learns to extract stable task-relevant cues rather than brittle lexical patterns. This significantly improves instruction-following reliability under natural linguistic variability commonly observed in real human–robot interaction.

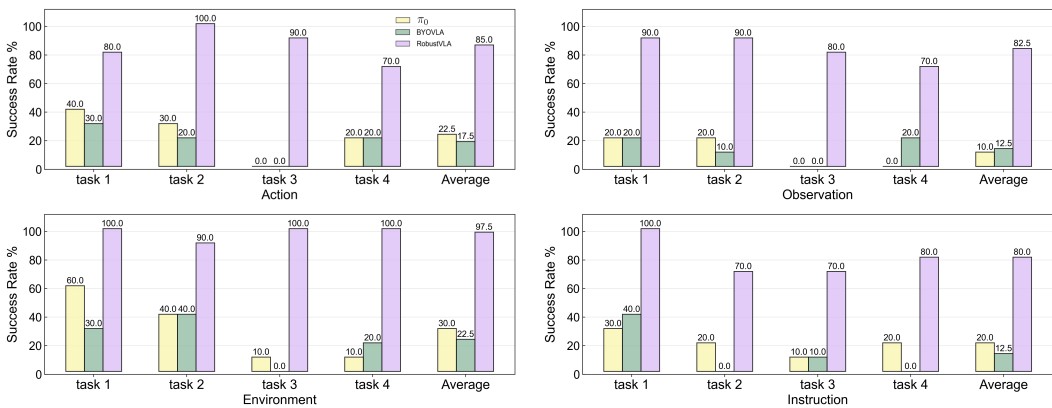

Figure 13: Details of Real-World Experimental Results

# D  PSEUDO CODE OF ROBUSTVLA

We present the pseudocode of our RobustVLA implemented on $\pi_0$ backbone in Algorithm. 1. Comparing with $\pi_0$, our RobustVLA gains robustness with minimal loss in clean performance by incorporating robustness against VLA input and output. In robustness against VLA input, we additionally use UCB algorithm to select the best perturbation. Both robustness against VLA input and output are trained using adversarial training with TRADES objective (Zhang et al., 2019), which optimally balance clean performance and robustness.

---

**Algorithm 1** Pseudo Code of RobustVLA Training on $\pi_0$

---

**Input:** Model $\theta$, dataset $\mathcal{D}$, augmentation set $\Omega$
**Output:** Trained model $\theta^*$
Initialize UCB balancer for $\Omega$
Initialize optimizer
**for** step $= 1$ to $T$ **do**
    Sample batch $\{(\mathbf{o}_t, A_t^1)\} \sim \mathcal{D}$
    *// UCB Step*
    Select augmentation $i^*$ using UCB
    $\omega^i(\mathbf{o}_t) \leftarrow \text{augment}(\mathbf{o}_t, i^*)$
    // Flow matching setup
    Sample $A_t^0 \sim \mathcal{N}(0, \mathbf{I})$, $\tau \sim \text{Beta}(1.5, 1)$
    $A_t^\tau \leftarrow \tau A_t^1 + (1 - \tau) A_t^0$
    $u \leftarrow A_t^0 - A_t^1$
    // Clean loss
    $v_\theta \leftarrow v_\theta(\mathbf{o}_t, A_t^\tau, \tau)$
    $\mathcal{L}_{\text{clean}} \leftarrow \|v_\theta - u\|^2$
    $\mathcal{L}_{\text{total}} \leftarrow \mathcal{L}_{\text{clean}}$
    *// Robust Against VLA Output*
    $\delta \leftarrow \text{random\_perturbation}$
    **for** $i = 1$ to $action\_pgd\_steps$ **do**
        $\mathcal{L}_{out} \leftarrow \max_{\|\delta\|_\infty \leq \epsilon_{\text{action}}} \mathbb{E}_{t,\epsilon} \left[ \left\| v_\theta \left( o_t, A_t^{\text{adv}}(\delta), t \right) - u_t^{\text{adv}}(\delta) \right\|^2 \right]$
        $\delta \leftarrow \text{PGD\_update}(\delta, \nabla_\delta \mathcal{L}_{out})$
    **end for**
    $\mathcal{L}_{total} \leftarrow \mathcal{L}_{total} + \mathcal{L}_{out}$
    *// Robust Against VLA Input*
    $\{\eta\} \leftarrow \text{random\_perturbation}$
    **for** $j = 1$ to $observation\_pgd\_steps$ **do**
        $\mathcal{L}_{in} \leftarrow \max_{\|\{\eta\}\|_\infty \leq \epsilon_{\text{obs}}} \mathbb{E}_{t,\epsilon} \left[ \left\| v_\theta \left( \omega^i(\mathbf{o}_t)^{\text{adv}}, x_t, t \right) - u_t \right\|^2 \right]$
        $\{\eta\} \leftarrow \text{PGD\_update}(\{\eta\}, \nabla_{\{\eta\}} \mathcal{L}_{in})$
    **end for**
    $\mathcal{L}_{total} \leftarrow \mathcal{L}_{total} + \mathcal{L}_{in}$
    Update $\theta$ with $\nabla_\theta \mathcal{L}_{\text{total}}$
    *// UCB update*
    Update UCB statistics with $-\mathcal{L}_{\text{in}}$
**end for**
**Return:** $\theta^* = \theta$

---

