# OpenReview forum: "On Robustness of Vision-Language-Action Model against Multi-Modal Perturbations"
_ICLR.cc/2026/Conference — ICLR 2026 Poster_

### Official Review · Reviewer_jsEV · 2025-10-20

**Soundness:** 2
**Presentation:** 2
**Contribution:** 3
**Rating:** 6
**Confidence:** 4

**Summary:**

This work evaluates the robustness of VLAs for robotic manipulation and identify the action modality as particularly fragile, in contrast to other modalities (vision, language, etc.) They propose RobustVLA, a framework designed to improve the robustness of VLAs to numerous axes of variation. The core of their method relies on adversarial training and a multi-armed banded (UCB) algorithm to select the most effective input perturbations. Results mostly include evaluations in the LIBERO benchmark, with a modest experimentation in the real-world. Their method achieves improvements relative to baseline models.

**Strengths:**

This paper's mathematical formulation of RobustVLA using the UCB algorithm appears novel and it is a creative application of the multi-armed bandit problem to VLAs. Moreover, the work considers an extensive amount of perturbations (17) and offers improvements in simulation (12% in LIBERO) and real. Moreover, the inference speed is low, which is valuable in robotic applications.

Additionally, the use of real-world experiments greatly strengthens the paper's claims because LIBERO and other simulation-based approaches often differ substantially from the real-world, limiting generalization of results. By having hardware experiments, there is great confidence in the results presented herein.

**Weaknesses:**

A minor concern is with the claim that $\pi_0$ is more robust with diffusion than an action-head; I don't think its tenable to make this claim with only a 5% increase in performance in LIBERO. The margin is too small and the sim-to-real gap is often large enough to make such claims unreliable. I suggest the author's revise this claim and discuss it with more nuance.

Additionally, while the real-world deployment of your method is commendable, the reliance on a small number of fine-tuning steps makes it difficult to assess the generalization of the results. When fine-tuning VLAs for a specific embodiment, often more than 25 demonstrations are required for good performance. While your approach may confer benefit in the low-data regime, it would be great to see how your method compares to the baselines as the number of demos increases, e.g., N=25, 50, 75, 100.

Finally, another point of contention is the use of the BYOVLA baseline. Upon reading that paper, the authors only use BYOVLA for visual robustness, but your experiments consider it for non-visual modalities as well? In this case, the claim that "existing visual-robust VLAs do not show improvements in other modalities" needs revises. I also would suggest framing prior work as a motivation, rather than a result, since BYOVLA never proposed evaluating other modalities such as language or actions.

I also would recommend clarifying that your approach fine-tunes a VLA, and is not a pre-training strategy when turning a VLM into a VLA.

Lastly, I would attempt to qualify some statements in Section 3, which only focus on simulation. There is a sim-to-real gap, and without extensive real-world experimentation, it is hard to confidently transfer results from one domain to the other.

**Questions:**

Clarification of the BYOVLA baseline: the manuscript states that it applies BYOVLA, a visual robustness method, to non-visual perturbations. However, little detail about the implementation is described. My primary concern stems from 1) possible misuse of BYOVLA for other modalities and 2) misuse of BYOVLA as a method. This work states they use a GradCAM (gradient-based attribution method) for BYOVLA but my reading of that paper indicates that the authors therein do not use GradCAM at all in their method. It would be great to provide a detailed description of how BYOVLA was applied to each modality since it was used as a baseline throughout the entire paper.

Rationale for small number of fine-tuning demonstrations: related to my previous point, 25 demos is often not enough for VLAs to adapt to a task. It would be great to see how performance of RobustVLA, and the other methods, scale as N increases from 25 up to 100-200. Its very surprising that a model like $\pi_0$ can't adapt to a FR5 setup.

Is omega time-invariant in Equation 1?

---

> ### Author Response · Authors · 2025-11-22
> **Response to Reviewer jsEV Part 1**
>
> We thank Reviewer jsEV for appreciating the novelty of the formulation, extensive perturbation analysis, value of real-world validation, and high inference efficiency of our paper. A detailed response is provided below.
>
> > Q1: A minor concern is with the claim that $\pi_0$ is more robust with diffusion than an autoregressive action-head; I don't think its tenable to make this claim with only a 5% increase in performance in LIBERO. The margin is too small and the sim-to-real gap is often large enough to make such claims unreliable. I suggest the author's revise this claim and discuss it with more nuance.
>
> Thanks for pointing this out. Indeed, current evidence is insufficient to verify this claim. To provide additional support, we add an additional experiment comparing $\pi_0$ and $\pi_0$-FAST with different magnitude of noise in action and observation. As shown in Appendix C.2, as noise increase, the performance of $\pi_0$-FAST experiences substantially faster degradation under perturbations in both modalities than $\pi_0$.
>
> However, we do admit that these evidence may not be enough to fully support the claim that diffusion-based action-head is more robust than its autoregressive counterpart. We have revised Section 3.2 in our main paper, our abstract and conclusion to pose this claim as a hypothesis for further researchers to verify, and express it in a less certain tone.
>
> > Q2: Additionally, while the real-world deployment of your method is commendable, the reliance on a small number of fine-tuning steps makes it difficult to assess the generalization of the results. When fine-tuning VLAs for a specific embodiment, often more than 25 demonstrations are required for good performance. While your approach may confer benefit in the low-data regime, it would be great to see how your method compares to the baselines as the number of demos increases, e.g., N=25, 50, 75, 100.
>
> Please allow us to first clarify that our fine-tuned $\pi_0$ achieves 100\% success rate with only 25 demonstrations, and we do not intentionally use less trajectories for fine-tuning.
>
> Following your advice, we evaluate our performance with 50 and 100 demonstrations to show results with sufficient data. Albeit all methods achieves 100\% success rate, $\pi_0$ gains additional robustness with more data. The success rate of $\pi_0$ increases from 37.5\% at 25 demos to 60.0\% at 50 demos and 65.0\% at 100 demos. This shows real-world robustness benefits from additional trajectories, which benefits the diversity of data during training. Comparing with baselines, our RobustVLA is highly robust in low-data regime, and consistently outperforms $\pi_0$ across all data scales. Our success rate reaches 92.5\% at 25 demos, 92.5\% at 50 demos and 95.0\% at 100 demos, while the baseline plateaus at 65.0\%, achieving 30\% robustness gain than $\pi_0$, showing RobustVLA adds robustness to non-robust models with diverse data only. These discussions are available in Line 517-525 in our main paper.
>
> Since this is a very important observation, we have updated our abstract, introduction and conclusion to reflect this. Our abstract is updated as follows: On the real-world FR5 robot, under four types of multimodal perturbations, RobustVLA shows strong low-data performance, achieving a 92.5\% success rate with only 25 demonstrations, compared to $\pi_0$ with 37.5\% success rate. Even with abundant demos, our method still outperform $\pi_0$ by 30\% success rate.

---

> ### Author Response · Authors · 2025-11-22
> **Response to Reviewer jsEV Part 2**
>
> > Q3: Finally, another point of contention is the use of the BYOVLA baseline. Upon reading that paper, the authors only use BYOVLA for visual robustness, but your experiments consider it for non-visual modalities as well? In this case, the claim that "existing visual-robust VLAs do not show improvements in other modalities" needs revises. I also would suggest framing prior work as a motivation, rather than a result, since BYOVLA never proposed evaluating other modalities such as language or actions. Also, BYOVLA never use Grad-CAM.
>
> Thanks for this advice. We extensively cite BYOVLA since it represents an important and meaningful milestone in improving VLA robustness. Technically, we use their original Github implementation of BYOVLA for visual modality only, with other modalities without robustness enhancement procedures. We have clarified this in Section 3.1 paragraph 1 to avoid misunderstanding.
>
> To avoid overclaim, we have also point out in Line 398 to show that our RobustVLA gains advantage over BYOVLA, and revised the text in Line 216-218 to indicate prior work as a motivation, rather than a result.
>
> We apologize for the mistake in the Grad-CAM issue. You are correct, the BYOVLA paper does not use Grad-CAM. In our implementation, we used the visual sensitivity probe proposed by BYOVLA correctly, but when preparing the manuscript we mistakenly described it as Grad-CAM. We apologize for this inaccuracy and have corrected the corresponding statements in Section 5.2. Importantly, the computational overhead we reported is measured using the correct BYOVLA probe in our codebase. Since the probe itself still requires high computation cost, our conclusions regarding computation cost remain unchanged.
>
> > Q4: Clarifying that your approach fine-tunes a VLA, and is not a pre-training strategy when turning a VLM into a VLA & qualifying some statements in Section 3
>
> Thanks for the suggestion. We have added explicit statements in Section 4 Line 1 to clarify this, which greatly improves our clarity.
>
> > Q5: Is $\omega$ time-invariant in Equation 1?
>
> During execution, the noise $\omega$ depends on the type of uncertainties. Time-variant disturbance includes Uniform Noise, External Force etc. which change dynamically during execution, while time-invariant disturbance includes instruction uncertainties (given at the beginning of the episode), action bias (added as a constant offset to simulate actuator error) etc.
>
> As for training time, although the 17 predefined perturbations used for evaluation remain fixed, the adversarial noises used during training, including the action noise $\delta$ and the observation noise $\eta$-are generated via PGD. These PGD-based perturbations evolve during training and continuously expose the weak spots of the current VLA model.

---

### Official Review · Reviewer_MSSU · 2025-10-24

**Soundness:** 2
**Presentation:** 3
**Contribution:** 2
**Rating:** 4
**Confidence:** 4

**Summary:**

This paper systematically studies the robustness of Vision–Language–Action (VLA) models under multimodal perturbations and proposes RobustVLA, a unified training framework to improve VLA robustness across four modalities: action, observation (vision), environment, and instruction. The authors first design a benchmark of 17 perturbations on the LIBERO suite and find that (1) the action modality is the most fragile, (2) existing vision-centric robustness methods do not generalize to other modalities, and (3) diffusion-based action heads (π₀) are inherently more robust than autoregressive ones.

**Strengths:**

1. 17 perturbations across four modalities, validated on two major VLA backbones and in real-world deployment.

2. The approach is ~50× faster in inference than some vision-robust baselines (e.g., BYOVLA), which is important for real-time robotics.

**Weaknesses:**

•	The use of flow-matching loss as a proxy for action quality is heuristic; correlation with actual task success is assumed but not analyzed quantitatively.

•	Performance dependence on ε, PGD steps, λ_in/out, and UCB α is not reported, raising reproducibility concerns.

•	It remains unclear how UCB avoids over-fitting to frequently selected perturbations; no ablation against uniform or curriculum sampling is shown.

•	The lack of domain randomization method: This method is the most common and effective method in the field of robot learning, but it is missing in this study.

**Questions:**

1.	Can you provide quantitative evidence that the flow-matching loss correlates with task success rate (e.g., correlation plots between loss change and success drop)?

2.	How is UCB initialized and tuned (α, exploration schedule)? Would uniform or curriculum sampling achieve similar results?

3.	How sensitive are the results to the adversarial ε and PGD steps?

4.	While some simple real-world experiments have been conducted, the 17 proposed perturbations have been extensively analyzed primarily in simulations. In real-world robotic manipulation tasks, which perturbations are the primary challenges that urgently need to be addressed, and which can be ignored? For example, the baseline algorithm pi0 can already handle long-horizon, two-arm clothes-folding tasks. Do these 17 perturbations actually exist in such real-world problems?

---

> ### Author Response · Authors · 2025-11-22
> **Response to Reviewer MSSU Part 1**
>
> We thank Reviewer MSSU for appreciating the scale of perturbations, validation on major VLA backbone, real-world experiment and fast inference speed of our paper. A detailed response is provided below.
>
> > Q1: The use of flow-matching loss as a proxy for action quality is heuristic; correlation with actual task success is assumed but not analyzed quantitatively.
>
> The use of flow-matching stems from multiple reasons. First, in adversarial attack community, attackers need a differentiable loss to compute gradient of the target network, which is used to generate the attack. The differentiable loss is then used as a proxy for the overall success. For example, in classification network, attackers to minimize the overall cross entropy loss, while the target is to minimize prediction accuracy [1]; in RL, attackers minimize the probability of taking the best action, while the target is to minimize task success rate [2]. Our use of flow-matching is motivated by its role as a differentiable loss that enables gradient-based perturbation generation.
>
> Second, for the task of VLA, obtaining a precise proxy for task success rate is hard since the control frequency is high, making credit assignment for each action step difficult, especially in VLA paradigms that do not assume assess to simulation environments. In this way, since VLA is essentially following an imitation learning paradigm, the learned policy with lower flow matching loss should be closer to the human demonstration trajectory and having higher probability for success, and intuitively, the policy having higher flow matching loss deviates from the demonstration, and may fail with a higher probability.
>
> Finally, following your suggestion, we have added an experiment to verify the connection between flow matching loss and task success rate. The results are added in Appendix C.5, which shows flow matching loss is strongly correlated with success rate, with significant Pearson correlation of $r=-0.95$, $p<0.05$.
>
> [1] Towards Deep Learning Models Resistant to Adversarial Attacks. ICLR 2018.
>
> [2] Adversarial attacks on neural network policies. 2017 ICLR workshop.
>
> > Q2: Performance dependence on $\epsilon$, PGD steps, $\lambda$\_in/out, and UCB $\alpha$ is not reported, raising reproducibility concerns. And How is UCB initialized and tuned?
>
> Please allow to clarify that these parameters are discussed in Section 5, Implementation details and full parameters at Appendix. B. Specifically, the $\epsilon$ is set to 0.03 for action space and 8/255 for observation space, PGD steps is set to 3 for faster training speed, $\lambda$\_in and $\lambda$\_out are set to 1, UCB $\alpha$ is set to 1. These hyperparameters are also available at our publicly released codebase. These hyperparameters are set using simple default values without extensive tuning, and works well empirically.
>
> We agree evaluating the performance under different hyperparameter initializations. As shown in Appendix B.2.1 and B.2.2, the performance of our RobustVLA is consistent with different PGD $\epsilon$, UCB exploration coefficient, UCB exponential moving average decay and UCB exponential moving average window, as long as the values are set reasonably.

---

> ### Author Response · Authors · 2025-11-22
> **Response to Reviewer MSSU Part 2**
>
> > Q3: It remains unclear how UCB avoids over-fitting to frequently selected perturbations; no ablation against uniform or curriculum sampling is shown.
>
> We answer this from the perspective of UCB itself and our implementation of UCB. In our paper, we formulate the problem of optimizing the overall robustness of VLAs as a multi-armed bandit, where each "arm" refers to a type of uncertainty to be explored. UCB algorithm is a principled way to solve multi-armed bandit with logarithmic regret. This is done by a UCB schedule of exploring the unpulled arms in the bandit and exploiting the pulled arms with high reward, balancing exploration-exploitation tradeoff.
>
> In our implementation of UCB, the reward for UCB is defined in Eqn. 7, which measures the vulnerability of VLA to each perturbation. As such, if VLA is more vulnerable to one perturbation, UCB receives high reward for pulling this perturbation "arm". As such, the UCB constantly select the most vulnerable perturbation for current model. As the model grows more robust, the reward gained by each perturbation is updated by exponential moving average, which guarantees each the reward gained by selecting each perturbation is maintained properly.
>
> Following your advice, we have also added an ablation of our RobustVLA without UCB, results are updated in Table 1, our main experiments. We also find Ours w/o UCB shows a 7.3\% drop in robustness comparing with RobustVLA, indicating that UCB is crucial for balancing multiple perturbations rather than letting the model overfit to a single dominant noise source. Additionally, comparing with Ours w/o in that trains on output noise only, adding exposure to diverse input noises adds further robustness gains, even without UCB. We have added this discussion in Line 415-421.
>
> [3] Using Confidence Bounds for Exploitation-Exploration Trade-offs. JMLR 2002.
>
> > Q4: While some simple real-world experiments have been conducted, the 17 proposed perturbations have been extensively analyzed primarily in simulations. In real-world robotic manipulation tasks, which perturbations are the primary challenges that urgently need to be addressed, and which can be ignored? For example, the baseline algorithm $\pi_0$ can already handle long-horizon, two-arm clothes-folding tasks. Do these 17 perturbations actually exist in such real-world problems?
>
> While $\pi_0$ can already handle long-horizon, two-arm clothes-folding tasks, many adaptations are required in practice. For example, when deploying $\pi_0$ to our Fairino FR5 robotic arm, it achieves 0% accuracy zero-shot. This is well-known as the cross-embodiment gap to VLAs, and greatly limit its real-world potential since a foundation model cannot be deployed directly, and still have to rely on expert demonstrations.
>
> As for the 17 perturbations, after consulting with engineers in several startups, they claim action perturbation to be the most common since they observe that a small variation on action can affect performance drastically. This may stem from initial calibration of robots or wear and tears after long time of operation. They also find environment and instruction uncertainties to be common in deployment, and observation uncertainties are observed occasionally.
>
> While we do not claim these 17 perturbations fully represents all types of uncertainties in real-world, our RobustVLA offers an offline approach for ensuring robustness against VLA input and output, which can be easily extended to industrial scenarios where engineers may be more familiar with the perturbations in their own cases.

---

> ### Author Response · Authors · 2025-11-22
> **Response to Reviewer MSSU Part 3**
>
> > Q5: The lack of domain randomization method: This method is the most common and effective method in the field of robot learning, but it is missing in this study.
>
> Please allow us to clarify that VLAs and domain randomization methods follow different learning paradigms. VLAs overcome the sim2real gap by collecting large amount of real-world data and do not rely on simulation completely. As a result, current VLAs like $\pi_0$ and OpenVLAs are learned via **offline learning**. Since all data represents real-world dynamics, there is in principle no issue of sim2real gap. In contrast, to the best of our knowledge, domain randomization methods assume the macro parameter of the simulation environment are bounded in a certain range, and the policy learned in such randomized environments can identify the maco parameters in real world and adapt to such environment dynamics. However, such paradigm assumes the existence of a simulator and is inherently **online learning** [4, 5].
>
> To mimic a domain randomization setting, we have added experiments in purely offline setting same as our RobustVLA, which trains $\pi_0$ using randomly selected VLA input perturbations and apply the same flow matching loss. The results are shown in Table 1, where DR performs poorly. While DR improves robustness to some observation noises, it fails under most environment and instruction noises, likely because it overfits to a small subset of easy perturbations. These discussions are added in Line 415-421.
>
> [4] Domain Randomization for Transferring Deep Neural Networks from Simulation to the Real World. IROS 2017.
>
> [5] A Bayesian Approach to Robust Reinforcement Learning. ICML 2020.

---

> ### Author Response · Authors · 2025-11-26
> **Invitation to reviewer-author discussion**
>
> Dear Reviewer,
>
> We are the authors of Submission 7027: "On Robustness of Vision-Language-Action Model against Multi-Modal Perturbations". We’ve uploaded a point-to-point response to your comments, which we believe addresses all of your concerns.
>
> As the reviewer-author discussion period is going to end, we would greatly appreciate it if you could take a moment to share your thoughts on our rebuttal. Your feedback is valuable to us, and we’d be happy to further clarify any remaining points if needed.
>
> Best regards, Authors of Submission 7027

---

### Official Review · Reviewer_w8cu · 2025-10-30

**Soundness:** 3
**Presentation:** 3
**Contribution:** 4
**Rating:** 6
**Confidence:** 3

**Summary:**

This paper investigates and enhances the robustness of Vision–Language–Action (VLA) models under multi-modal perturbations spanning actions, language instructions, environments, and visual observations. The study finds that action perturbations are the most fragile modality, existing visual-robust VLAs fail to generalize beyond visual noise, and the π₀ model, equipped with a diffusion-based action head, exhibits the strongest inherent robustness.

To address these weaknesses, the authors introduce RobustVLA, a unified framework that strengthens robustness through two key components:
Output robustness: adversarially optimzing against worst-case action noise via a flow-matching objective;
Input robustness: enforcing action consistency under semantically equivalent input variations, with a UCB-based adaptive mechanism to prioritize the most harmful perturbations during training.

Extensive experiments on the LIBERO benchmark demonstrate that RobustVLA achieves +12.6% robustness on π₀ and +10.4% on OpenVLA, while maintaining over 50× faster inference compared to prior visual-robust models. In real-world robotic evaluations, RobustVLA further improves success rates by 65.6% under combined multi-modal disturbances.

**Strengths:**

- The paper presents a comprehensive evaluation of Vision–Language–Action (VLA) models under 17 perturbations across four modalities, offering a detailed analysis of their robustness weaknesses.
- The proposed RobustVLA framework effectively improves resilience to both input and output perturbations through offline optimization against worst-case action noise and a multi-armed bandit–based adaptive training strategy.
- RobustVLA achieves significant robustness gains while maintaining strong performance under clean, unperturbed conditions.
- The framework demonstrates strong real-world generalization on the FR5 robot, delivering a 65.6% performance improvement under multi-modal disturbances with only limited demonstrations.
- RobustVLA also offers high computational efficiency, achieving faster inference than prior visual-robust VLAs and eliminating reliance on external large models, making it well-suited for practical robotic applications.

**Weaknesses:**

- The real-world evaluation is limited to a small set of tasks conducted on a single robotic platform (FR5). Broader validation across multiple robots, environments, and task domains would strengthen the claims of generalizability.
- The RobustVLA framework integrates several advanced components. Although each element is clearly described, the overall methodological complexity may hinder ease of adoption and reproducibility for researchers less familiar with these techniques.
- The paper touches on the robustness–accuracy trade-off but lacks a detailed quantitative or theoretical analysis of this balance. A deeper exploration of how robustness affects clean performance across varying tasks and noise levels would provide valuable insight.

**Questions:**

The real-world experiments were conducted on a single robotic platform (FR5 robot). ​ Can the authors provide insights into how RobustVLA would perform on other robotic platforms or in different environments? Are there any specific limitations or challenges in adapting the framework to other systems?

The paper states that hyperparameters for UCB exploration and adversarial training were not extensively optimized. ​ Could the authors provide more details on how sensitive the framework is to these hyperparameters? Are there plans to explore optimization techniques for these parameters?

The failure analysis in the real-world experiments is insightful but could benefit from more detailed explanations of why the baselines failed under specific perturbations. This would help readers better understand the unique advantages of RobustVLA.

---

> ### Author Response · Authors · 2025-11-22
> **Response to Reviewer w8cu Part 1**
>
> We thank Reviewer w8cu for appreciating the comprehensive  robustness evaluation, effectiveness of our method, significant robustness gain, strong real-world performance and high computational efficiency of our paper. A detailed response is provided below.
>
> > Q1: The real-world evaluation is limited to a small set of tasks conducted on a single robotic platform (FR5). Broader validation across multiple robots, environments, and task domains would strengthen the claims of generalizability.
>
> Thanks for your helpful suggestion. Due to the limited assess to infrastructures, we have to conduct our major study in simulation and evaluate real-world performance on FR5 platform. We consider evaluation on other robot platforms as our future work when more funding support is available.
>
> > Q2: The RobustVLA framework integrates several advanced components. Although each element is clearly described, the overall methodological complexity may hinder ease of adoption and reproducibility for researchers less familiar with these techniques.
>
> Please allow us to clarify that robustness against multi-modal uncertainties inherently require robustness against VLA input and output. Our method draws inspiration in adversarial robustness literature and adapt it to VLA architectures, where our input and output robustness objective is a modified version of rectified flow matching objective in $\pi_0$. The code below shows our core modifications of our $\mathcal L_{out}$ loss, which is only 5 lines of code.
>
> pi0.py:
> ```python
> noise = random_normal_like(actions)
> time = sample_beta(batch)
> x_t = time * noise + (1 - time) * actions
> u_t = noise - actions
>
> v_t = model_forward(obs, x_t, time)
> loss_clean = mse(v_t, u_t)
> loss_total = loss_clean
> ```
>
> pi0_act_adv.py:
> ```diff
> noise = random_normal_like(actions)
> time = sample_beta(batch)
> x_t = time * noise + (1 - time) * actions
> u_t = noise - actions
>
> v_t = model_forward(obs, x_t, time)
> loss_clean = mse(v_t, u_t)
> loss_total = loss_clean
>
> + eta = uniform_like(actions, -eps, eps)
> + eta = eta + alpha * sign(grad(lambda e: mse(model_forward(obs, x_t_adv(e)), u_t), eta))
> + x_t_adv = time * noise + (1 - time) * (actions + eta)
> + loss_adv_act = mse(model_forward(obs, x_t_adv, time), u_t)
> + loss_total = loss_clean + loss_adv_act
> ```
>
> > Q3: The paper touches on the robustness–accuracy trade-off but lacks a detailed quantitative or theoretical analysis of this balance. A deeper exploration of how robustness affects clean performance across varying tasks and noise levels would provide valuable insight.
>
> The decrease in clean accuracy when optimizing our RobustVLA objective is theoretically inevitable since apart from the raw $\pi_0$ loss, we are additionally optimizing a robust loss that shifts the decision boundary. This is illustrated in the TRADES paper [1], as we have cited in line 261.
>
> Specifically, TRADES shows that optimizing robustness includes a natural error (flow matching loss without noise) and a boundary error term (VLA under perturbations), the boundary error term enforce local invariance of VLAs to adversarial perturbations, which increases natural error when the data are not separable with margin greater than the perturbation boundary. This establishes a fundamental trade-off: improving robustness requires enlarging perturbation margins around clean trajectories, which shifts the decision boundary and can reduce clean flow matching performance. The hyperparameter $\lambda$\_in and $\lambda$\_out in our objective explicitly controls this balance through this principled tradeoff. Therefore, the observed reduction in clean accuracy is consistent with the theoretical characterization of the accuracy–robustness trade-off and is not specific to our implementation.
>
> Beyond theoretical insight, TRADES also enables optimal robustness-accuracy tradeoff, resulting in large robustness gains with minimal accuracy degradation. This is done by penalizing local prediction inconsistency rather than enforcing adversarially-invariant labels, which they theoretically guarantee that no surrogate based only on local neighborhood consistency can be tighter (ie, achieving better accuracy-robustness tradeoff) than their objective. As such, our RobustVLA inherits this benefit from TRADES and empirically gain large robustness with minimal degradation in accuracy, as verified by our simulation results in LIBERO benchmark. In real world, our RobustVLA 100\% success rate without perturbations, demonstrating our RobustVLA does not harm clean performance.
>
> Additionally, following your advice, we have evaluated the performance between varying noise levels in Appendix C.4, our RobustVLA consistently outperforms the baselines, with the gain becoming more pronounced as the noise level increases.
>
> [1] Theoretically principled trade-off between robustness and accuracy. ICML 2019.

---

> ### Author Response · Authors · 2025-11-22
> **Response to Reviewer w8cu Part 2**
>
> > Q4: hyperparameters for UCB exploration and adversarial training
>
> Following your advice, we have tested the performance under UCB hyperparameters and adversarial training hyperparameters, and the result remains stable. The performance is shown in Appendix B.2.2. We find the performance relatively stable with different exploration coefficient, exponential moving average decay and window size. We find the performance consistent across all settings, showing our UCB is not sensitive to these variants when the hyperparameters are relatively reasonable.
>
> As for adversarial training, we have added ablations on different PGD $\epsilon$ in Appendix B.2.1. The performance  remains stable with $\epsilon$=0.015, 0.03 and 0.06, all largely outperforming the $\pi_0$ baseline, showing our method is not sensitive with PGD $\epsilon$.
>
>
> > Q5: more detailed explanations of why the baselines failed under specific perturbations
>
> Thank you for the suggestion. We have added additional analysis in Appendix C.6. We summarize the main points below:
>
> **Action Uncertainty.**
> Baseline VLAs collapse under even small action noise because imitation-trained policies lack recovery behaviors. Once an action deviates, errors accumulate and the policy cannot self-correct. This leads to pose drift and oscillatory motions in real world. RobustVLA avoids this brittleness by adversarially training against worst-case action perturbations, smoothing the action manifold and preventing single-step failures.
>
> **Observation Uncertainty.**
> Visual corruptions distort object boundaries and task states, causing baselines to misinterpret progress (e.g., repeated grasps after the task is complete). BYOVLA helps with static distractions but fails under dynamic or semantic corruption. RobustVLA enforces consistent actions under semantically preserving perturbations and prioritizes harmful corruptions via UCB, greatly improving robustness to noisy or degraded observations.
>
> **Environment Uncertainty.**
> Scene changes (distractors, lighting, layout differences) push baselines OOD, leading to mis-tracked objects or unsafe motions due to overfitting to narrow demonstration distributions. RobustVLA repeatedly trains on perturbations that maximally increase loss, improving generalization to unseen layouts and dynamic environments.
>
> **Instruction Uncertainty.**
> Baselines rely on brittle keyword cues, so synonym substitutions or reordered phrasing often change their behavior. RobustVLA treats semantically equivalent linguistic variants as invariances, encouraging stable grounding and yielding reliable execution under natural language variability.

---

### Official Review · Reviewer_Q9r5 · 2025-11-02

**Soundness:** 3
**Presentation:** 3
**Contribution:** 4
**Rating:** 6
**Confidence:** 4

**Summary:**

This paper tackles the challenge of multi-modal robustness in Vision-Language-Action (VLA) models, which often fail under real-world noise beyond visual corruption. Through systematic evaluation of leading VLAs (OpenVLA, π₀, BYOVLA) across 17 perturbations in four modalities (action, observation, environment, and instruction), the authors find that actions are the most fragile, visual-robust methods don’t generalize to other modalities, and diffusion-based π₀ is inherently more robust than autoregressive models.

To overcome these issues, they propose RobustVLA, a framework that enhances robustness for both outputs (actions) and inputs (visual, language, and environmental signals). It adversarially trains against worst-case action noise in the flow-matching objective and enforces consistent actions across perturbed inputs using an adaptive UCB-based noise selection strategy.

On the LIBERO benchmark, RobustVLA improves robustness by 12.6% on π₀ and 10.4% on OpenVLA, while being 50× faster than BYOVLA. In real-world robot tests, it achieves 65.6% higher success under diverse perturbations. Overall, RobustVLA provides an efficient, unified solution for building robust and reliable VLA models across all modalities.

**Strengths:**

1. Comprehensive multi-modal robustness evaluation is very important for the VLA research. The paper goes beyond the common focus on visual perturbations and systematically evaluates 17 types of noise across four modalities (action, observation, environment, and instruction). This provides one of the most complete analyses of VLA robustness to date and clearly identifies action perturbations as the dominant failure mode

2. RobustVLA demonstrates consistent and significant gains in robustness across all tested modalities, with +12–13% improvement in simulation and +65% in real-world robot experiments, while maintaining clean performance and 50× faster inference compared to visual-robust baseline models.

3. The framework generalizes to both diffusion-based and autoregressive VLAs (π₀ and OpenVLA) and does not rely on external large models or expensive perception modules.

4. The paper is well-structured, clearly presents its findings.

**Weaknesses:**

1. Most experiments only cover a small set of tabletop manipulation tasks with relatively simple perturbations. The results may not fully demonstrate robustness under more complex, long-horizon, or dynamic real-world scenarios. It is unclear whether different levels of perturbations may have various performance observations.

2. Discussion on trade-offs and generalization: The paper claims robustness without loss of clean accuracy, but does not explore possible trade-offs in data efficiency, stability during training, or transferability to new robot morphologies or unseen tasks. These aspects are important for assessing the general utility of RobustVLA.

3. Table 1 provides results on 17 noise types, yet the analysis mostly reports average success rates without detailed per-modality insight. For example, under observation noise, RobustVLA’s improvement on “Dead Pixel” is dramatic (20.8 → 93.8%), but for “Color Jitter,” the gain is moderate (61.7 → 69.5%). The paper does not explain why certain corruptions benefit more or how robustness scales with noise level (beyond the limited Figure 3a). A deeper breakdown, e.g., visualizing robustness curves per modality or ablation across perturbation strength, would better reveal the method’s internal behavior.

**Questions:**

While the paper reports similar clean performance to π₀ (95.5% vs. 96.0%), it remains uncertain whether the robustness-oriented training objective might negatively affect performance on other standard or real-world benchmarks not included in LIBERO. Since real-world policies often require precise control in unperturbed environments, it would be valuable to evaluate whether RobustVLA maintains its performance on such original tasks without introducing unintended degradation. Could authors provide more explanations on it?

---

> ### Author Response · Authors · 2025-11-22
> **Response to Reviewer Q9r5 Part 1**
>
> We thank Reviewer Q9r5 for appreciating the novelty of multi-modal evaluation, consistent and significant gains in robustness, generalization capability of technical framework and clarity of our paper. A detailed response is provided below.
>
> > Q1: Most experiments only cover a small set of tabletop manipulation tasks with relatively simple perturbations. The results may not fully demonstrate robustness under more complex, long-horizon, or dynamic real-world scenarios. It is unclear whether different levels of perturbations may have various performance observations.
>
> As for more complex, long-horizon and dynamic scenarios, our LIBERO simulation benchmark includes LIBERO-long subtask, which focus on long-horizon control. As shown in Appendix C.3, RobustVLA consistently improves over $\pi_0$ on all LIBERO-long tasks (average +19.61\% success rate). This supports the applicability of our method to long-horizon control.
>
> We additionally evaluate robustness under varying noise levels for action and observation modalities. As shown in Appendix C.4, our RobustVLA consistently outperforms the baselines, with the gain becoming more pronounced as the noise level increases. We believe the experiments above verify the effectiveness of our RobustVLA methods.
>
>
> > Q2: Discussion on trade-offs and generalization: The paper claims robustness without loss of clean accuracy, but does not explore possible trade-offs in data efficiency, stability during training, or transferability to new robot morphologies or unseen tasks. These aspects are important for assessing the general utility of RobustVLA.
>
> Thanks for raising this important point. While our primary focus is on robustness of VLA against multi-modal uncertainties, we acknowledge these points are important for VLA deployment. We added an experiment on the generalization of our RobustVLA by training $\pi_0$ and RobustVLA using 25, 50 and 100 demonstration trajectories in real world. Albeit all methods achieves 100\% success rate, $\pi_0$ gains additional robustness with more data. The success rate of $\pi_0$ increases from 37.5\% at 25 demos to 60.0\% at 50 demos and 65.0\% at 100 demos. This shows real-world robustness benefits from additional trajectories, which benefits the diversity of data during training. Comparing with baselines, our RobustVLA consistently gains robustness in low-data regime, and gains additional robustness with more data. Our success rate reaches 92.5\% at 25 demos, 92.5\% at 50 demos and 95.0\% at 100 demos, while the baseline plateaus at 65.0\%, achieving 30\% robustness gain than $\pi_0$. The result shows our RobustVLA gains additional robustness than adding training data diversity alone. We have added this in Line 514-523, and correct our main statements in abstract, introduction and conclusion to reflect this.
>
> For training stability, we additionally include the training-loss curve in Appendix B.3, which demonstrates smooth and stable convergence under our RobustVLA objective, indicating that the learning process converges efficiently. We have also provided some suggestions on tuning VLAs for future practitioners to use our code more efficiently.

---

> ### Author Response · Authors · 2025-11-22
> **Response to Reviewer Q9r5 Part 2**
>
> > Q3: Table 1 provides results on 17 noise types, yet the analysis mostly reports average success rates without detailed per-modality insight. For example, under observation noise, RobustVLA’s improvement on “Dead Pixel” is dramatic (20.8 → 93.8%), but for “Color Jitter,” the gain is moderate (61.7 → 69.5%). The paper does not explain why certain corruptions benefit more or how robustness scales with noise level (beyond the limited Figure 3a). A deeper breakdown, e.g., visualizing robustness curves per modality or ablation across perturbation strength, would better reveal the method’s internal behavior.
>
> Mathematically, we classify our noise types as noise in VLA input (observation, instruction and environment), and noise in VLA output (action), as they represent different elements in MDPs. As such, we find actions the hardest noise to defend and input noises relatively easy. Note that the external force in environment uncertainty is also applied on action. As such, our experiment result shows RobustVLA achieves +8.0\% success rate to uncertainties in VLA output on average, comparing with +17.2\% increase in uncertainties applied to VLA input. This results again echos the findings in Section 3.2, showing action the most fragile modality.
>
> To further verify this, we have additionally added an experiment to test RobustVLA under different noise levels in Appendix C.4. Under action modality, the average imporvement is 5.6\% and under observation modality, the average improvement is 23\%, showing noises applied to action is hard to defend, and we highlight this as an open problem.
>
> As for the moderate gain in color jitter, we conducted a pilot experiment and find this noise is intrinsically hard to defend, while training specifically against it even harm overall robustness against other noises. As such, model optimzing overall performance may pay less attention on this "hard" example. We also find our UCB having high chance of selecting color jitter at the end of training (43.73\%), but the performance do not improve.
>
> > Q4: While the paper reports similar clean performance to $\pi_0$ (95.5% vs. 96.0%), it remains uncertain whether the robustness-oriented training objective might negatively affect performance on other standard or real-world benchmarks not included in LIBERO. Since real-world policies often require precise control in unperturbed environments, it would be valuable to evaluate whether RobustVLA maintains its performance on such original tasks without introducing unintended degradation. Could authors provide more explanations on it?
>
> The decrease in clean accuracy when optimizing our RobustVLA objective is theoretically inevitable since apart from the raw $\pi_0$ loss, we are additionally optimizing a robust loss that shifts the decision boundary. This is illustrated in the TRADES paper [1], as we have cited in line 274.
>
> Specifically, in TRADES, they proves that optimizing robustness includes a natural error (flow matching loss without noise) and a boundary error term (VLA under perturbations), the boundary error term enforce local invariance of VLAs to adversarial perturbations, which increases natural error when the data are not separable with margin greater than the perturbation boundary. This establishes a fundamental trade-off: improving robustness requires enlarging perturbation margins around clean trajectories, which shifts the decision boundary and can reduce clean flow matching performance. The hyperparameter $\lambda$\_in and $\lambda$\_out in our objective explicitly controls this balance through this principled tradeoff. Therefore, the observed reduction in clean accuracy is consistent with the theoretical characterization of the accuracy–robustness trade-off and is not specific to our implementation.
>
> Beyond theoretical insight, TRADES also enables optimal robustness-accuracy tradeoff, resulting in large robustness gains with minimal accuracy degradation. This is done by penalizing local prediction inconsistency rather than enforcing adversarially-invariant labels, which they theoretically guarantee that no surrogate based only on local neighborhood consistency can be tighter (ie, achieving better accuracy-robustness tradeoff) than their objective. As such, our RobustVLA inherits this benefit from TRADES and empirically gain large robustness with minimal degradation in accuracy, as verified by our simulation results in LIBERO benchmark. In real world, our RobustVLA 100\% success rate without perturbations, demonstrating our RobustVLA does not harm clean performance.
>
> [1] Theoretically principled trade-off between robustness and accuracy. ICML 2019.

---

### Official Review · Reviewer_hH7B · 2025-11-02

**Soundness:** 4
**Presentation:** 2
**Contribution:** 3
**Rating:** 8
**Confidence:** 4

**Summary:**

This paper simply run a study and find analyze the robustness of multiple policy models agains 17 different perturbations and finds that the action perturbations have the most impact while Pi-0 model is the most robust one.

They propose a model, based on Pi-0, for a robust policy against all modalities perturbations. Authors show significantly better robustness for the proposed method while the success rate is also best or close to the best. The model contains 3 loss terms, one is based on pi-0, second adds robustness agains semantically preserving input perturbation (KL divergence) and the third one adds robustness against output perturbations (adversarial worst case perturbation).

**Strengths:**

Superior results on both task accuracy and also perturbation resilience.

Study before formulation. I enjoyed the first study and findings on different methods and types of perturbations. It made the intuition behind the proposed model design clear.

**Weaknesses:**

Please read my questions section.

**Questions:**

1- The worst perturbation, computed by PGD, is a theoretical worst case and not necessarily represents the robot/environment real world noise/issues distribution. How is this justified in your proposed formulation?

2- How come the noise accumulation in series of actions is addressed in this formulation? I assume smallest noise in each step will be propagated and accumulated with previous and next time-steps noise.

3- Rectified flow makes the modeling and compute easier, but I am not sure how it is a valid assumption for this problem. Why can we assume a constant velocity for all the steps?

---

> ### Author Response · Authors · 2025-11-22
> **Response to Reviewer hH7B**
>
> We thank Reviewer hH7B for appreciating the motivation, superior experiment result and clarity of our paper. A detailed response is provided below.
>
> > Q1: The worst perturbation, computed by PGD, is a theoretical worst case and not necessarily represents the robot/environment real world noise/issues distribution. How is this justified in your proposed formulation?
>
> In the literature of adversarial robustness, it is common to assume the uncertainties are bounded by some constraints, such that the perturbation do not drift from the clean training dataset too far. In this way, by assuming bounded worst-case perturbation, we are maximizing the robustness lower bound in the perturbation set, thus automatically guaranteed robustness against real-world perturbations in the perturbation set we optimize.
>
> For noises outside the PGD perturbation set, it is empirically shown that enhancing adversarial robustness in $\ell_p$ ball generalize to other forms of perturbations [1-2]. Empirically, as also shown in our paper, while our adversarial PGD is $\ell_p$ bounded, many of the perturbations we evaluate fall outside this $\ell_p$ ball, yet RobustVLA still achieves consistent robustness improvements. This is especially evident in *external force* perturbation, where we add a large, abrupt foce to the robot, while our RobustVLA gains robustness despite never trained on such cases. We have added this discussion in Line 401-402.
>
> [1] Adversarial Training and Robustness for Multiple Perturbations. NeurIPS 2019.
>
> [2] Towards Compositional Adversarial Robustness: Generalizing Adversarial Training to Composite Semantic Perturbations. CVPR 2023.
>
>
> > Q2: How come the noise accumulation in series of actions is addressed in this formulation? I assume smallest noise in each step will be propagated and accumulated with previous and next time-steps noise.
>
> Indeed, noise accumulation accounts for the non-robustness in action modality, since the non-optimal perturbed action accumulates gradually, driving policy to out-of-distribution states [1]. This is addressed in our Section 4.1, Robustness against VLA outputs. Specifically, our objective can be explained as: (1) flow matching against both clean and perturbed action distributions, improving closed-loop stability under test-time noise. As a result, the flow matching process against action noise explicitly constrain the stepwise deviation in action distributions, preventing action noise to accumulate. (2) penalizing outliers that accumulates action noise largely. Our attack can be explained as maximizing a MSE loss using a worst-case action perturbation $\delta$. As such, if a noise can drastically drive flow matching to its failure, our loss will penalize such noise quadratically, limiting the noise accumulation process. Please refer to our Remark 1-3 in Line 272-296 for these discussions.
>
> [3] Offline reinforcement learning: Tutorial, review, and perspectives on open problems. 2020.
>
> > Q3: Rectified flow makes the modeling and compute easier, but I am not sure how it is a valid assumption for this problem. Why can we assume a constant velocity for all the steps?
>
> Please allow us to clarify that our method in Section 4.1 do not assume rectified flow as an necessary condition. To show this clearly, we have revised Line 229-232 to show we take $\pi_0$ with rectified flow matching as an example due to its popularity, while our method naturally extends to VLAs with general diffusion-based action head and autoregressive VLAs like OpenVLA. We also add an explicit "Generalizing to Other VLAs" paragraph in Line 296-302 and Line 351-353 to show how our method generalize to general diffusion-based action head and autoregressive VLAs.
>
> To explain, in Eqn. 3, the worst-case action noise is primarily defined as:
> $$
> \delta \in \text{argmax}_\delta \mathbb E _{p(A_t^1 \mid \mathbf o _t), q(\hat{A}_t^\tau \mid \hat{A}_t^1)} || v _{\theta}(\hat{A}_t^\tau, \mathbf o_t) - u(\hat{A}_t^\tau \mid \hat{A}_t^1) ||^2,
> $$
> which can be computed by PGD and do not assume rectified flow. Here, $\hat{A}_t^\tau$ can be computed by any diffusion algorithm such as linear Gaussian. The action noise $\delta$ can then be used for adversarial training in Eqn. 4.
>
> However, we would like to show using the assumption of rectified flow provides additional theoretical insights. In Eqn. 3, the second equation assumes rectified flow $u(\hat{A}_{t} ^\tau|\hat{A}_t^1) = u(A_t^\tau|A_t^1) - \delta$ and gets:
>
> $$
> \text{argmax}_\delta \mathbb E _{p(A_t^1 \mid \mathbf o _t), q(\hat{A}_t^\tau \mid \hat{A}_t^1)} || v _{\theta}(\hat{A}_t^\tau, \mathbf o_t) - u(A_t^\tau | A_t^1) - \delta ||^2 .
> $$
>
> Since $u(\hat{A}_{t} ^\tau|\hat{A}_t^1) = u(A_t^\tau|A_t^1) - \delta$ , we know the noise is in the direction of $v _\theta(\hat{A}_t^\tau, \mathbf o_t) - u(A_t^\tau | A_t^1)$, where velocity field and rectified flow diverge the most, making readers easier to understand.

---

### Meta-Review · Area_Chair_rZcF · 2026-01-09

**Summary:**

This paper presents a study of robustness in Vision-Language-Action (VLA) models under 17 perturbations across four modalities, and introduces RobustVLA, a unified framework to improve robustness to both input and output uncertainties. Reviewers found the problem important, appreciated the robustness evaluation, and found the empirical results in both simulation and real-world robotic experiments valuable. The proposed method achieves robustness gains while remaining computationally efficient, which was highlighted as a key strength.

**Reviewer Concerns:**

Addressed:

- The authors provided quantitative evidence linking flow-matching loss to task success.

- Ablations and sensitivity analyses were added for adversarial training, UCB selection, and hyperparameters.

- Clarifications were made regarding the role of diffusion and robustness–accuracy trade-offs.

- Claims about diffusion robustness and BYOVLA were revised and corrected.

-Real-world evaluations were expanded across multiple data regimes, strengthening generalization claims.


Outstanding:

- Real-world validation is still limited to a single robotic platform and a small set of tasks.

- Some concerns remain about broader generalization to more complex, long-horizon, or diverse real-world settings.

- Methodological complexity may limit ease of adoption for some practitioners.

**Reviewer Scores:**

The reviewers scores would likely remain unchanged (positive).
Reviewer MSSU might slightly improve (from marginally below the acceptance threshold to  marginally above the acceptance threshold)

---

### Decision · Program_Chairs · 2026-01-26

Accept (Poster)